# Robust charge-density wave strengthened by electron correlations in monolayer 1T-TaSe$_2$ and 1T-NbSe$_2$

Yuki Nakata[1], Katsuaki Sugawara [1,2,3], Ashish Chainani[4], Hirofumi Oka [3], Changhua Bao[5], Shaohua Zhou[5], Pei-Yu Chuang[4], Cheng-Maw Cheng [4], Tappei Kawakami[1], Yasuaki Saruta[1], Tomoteru Fukumura[6], Shuyun Zhou [5,7], Takashi Takahashi[1,2,3] & Takafumi Sato [1,2,3 ✉]

Combination of low-dimensionality and electron correlation is vital for exotic quantum phenomena such as the Mott-insulating phase and high-temperature superconductivity. Transition-metal dichalcogenide (TMD) 1T-TaS$_2$ has evoked great interest owing to its unique nonmagnetic Mott-insulator nature coupled with a charge-density-wave (CDW). To functionalize such a complex phase, it is essential to enhance the CDW-Mott transition temperature $T_{\text{CDW-Mott}}$, whereas this was difficult for bulk TMDs with $T_{\text{CDW-Mott}} < 200$ K. Here we report a strong-coupling 2D CDW-Mott phase with a transition temperature onset of ~530 K in monolayer 1T-TaSe$_2$. Furthermore, the electron correlation derived lower Hubbard band survives under external perturbations such as carrier doping and photoexcitation, in contrast to the bulk counterpart. The enhanced Mott-Hubbard and CDW gaps for monolayer TaSe$_2$ compared to NbSe$_2$, originating in the lattice distortion assisted by strengthened correlations and disappearance of interlayer hopping, suggest stabilization of a likely nonmagnetic CDW-Mott insulator phase well above the room temperature. The present result lays the foundation for realizing monolayer CDW-Mott insulator based devices operating at room temperature.

[1] Department of Physics, Graduate School of Science, Tohoku University, Sendai 980-8578, Japan. [2] Center for Spintronics Research Network, Tohoku University, Sendai 980-8577, Japan. [3] Advanced Institute for Materials Research (WPI-AIMR), Tohoku University, Sendai 980-8577, Japan. [4] National Synchrotron Radiation Research Center, Hshinchu 30077, Taiwan ROC. [5] State Key Laboratory of Low Dimensional Quantum Physics and Department of Physics, Tsinghua University, Beijing 100084, China. [6] Department of Chemistry, Graduate School of Science, Tohoku University, Sendai 980-8578, Japan. [7] Frontier Science Center for Quantum Information, Beijing 100084, China. ✉email: t-sato@arpes.phys.tohoku.ac.jp

The interplay among electron correlation, dimensionality, and appearance of various quantum phases is a long-standing issue in condensed-matter physics. The correlated electron system is characterized by strong Coulomb interactions among electrons and the resultant emergence of exotic physical properties, which are absent in the weakly interacting electron system. The most drastic consequence of electron correlation is the typical Mott–Hubbard transition[1,2] that converts a half-filled paramagnetic metal (predicted by single-particle theory) into an antiferromagnetic insulator when the on-site Coulomb interaction $U$ exceeds the bandwidth $W$ (i.e., effective Coulomb interaction $U/W > 1$). A more unusual phase is the nonmagnetic Mott insulator and associated exotic quantum phases, as highlighted by the quantum-spin-liquid phase in 1T-TaS$_2$, a triangular lattice of two-dimensional (2D) Mott insulator with a CDW[3,4]. In comparison, the destruction of antiferromagnetic order in doped copper oxides leads to emergence of high-temperature superconductivity that coexists with charge order[5,6].

To realize a Mott insulator, it is essential that the magnitude of $U/W$ is above a critical value relevant to the structure and the electronic states of a material[1,2]. In fact, for an optimally doped copper oxide, the large $U$ (~3 eV)[7] estimated for Cu-3$d$ electrons compared with the relatively small bandwidth $W$ (~0.4 eV)[8] also satisfies the condition of $U/W \gg 1$. This suggests a direct relation between Mott physics and superconductivity (note that, in cuprates, the role of $U$ is actually played by the charge-transfer gap of 1.4–2.0 eV[9], but even in this case, the effective $U/W$ (3.5–5.0) exceeds the critical value). However, the recent discovery of a Mott-insulator phase and associated superconductivity in tilted bilayer graphene[10,11] demonstrated that even when $U$ is considerably small (~30 meV), the band narrowing ($W \sim 20$ meV) introduced by the superstructure of moiré pattern can effectively convert a metallic state into a Mott-insulating one. This points to the importance of bandwidth control for materials with small $U$ to trigger the Mott transition.

The layered transition-metal dichalcogenide (TMD) 1T-TaS$_2$ is believed to be a special example of a bandwidth-controlled Mott-transition material[12,13] in the absence of magnetic order. Bulk 1T-TaS$_2$ undergoes a Mott transition accompanying a commensurate charge-density wave (CDW) characterized by the star-of-David cluster (Fig. 1a) with a $\sqrt{13} \times \sqrt{13}$ periodicity (Fig. 1b), at $T_{\text{CDW-Mott}} \sim 200$ K. It is noted that twelve Ta atoms located at the corners of a cluster are slightly displaced from the original position toward the central Ta atom (Fig. 1a). 1T-TaS$_2$ satisfies the half-filling condition necessary for realizing a Mott-insulator phase, since 12 electrons at the displaced 12 Ta atoms form the fully occupied 6 bands and the remaining electron at the central Ta atom forms a half-filled metallic band[14,15]. Although the $U$ of Ta 5$d$ electrons is relatively small (~0.7 eV)[16], 1T-TaS$_2$ undergoes the Mott transition when the half-filled band is narrowed to the scale of $U$ due to the band folding associated with the CDW[12,13] in a similar manner to tilted bilayer graphene. More interestingly, it was shown that while it has a charge gap of ~0.3 eV, it shows gapless quantum-spin-liquid dynamics and no long-range magnetic order down to 70 mK[17]. Recently, the exploration for Mott phases coexisting with CDW was extended to the atomic-layer limit in TMDs as in graphite (graphene), with the possible emergence of exotic quantum phenomena in the pure 2D limit[18–20]. However, the nature of a pure 2D CDW-Mott phase, such as its robustness, possibility for magnetism, and differences if any, compared with the 3D bulk case, has been scarcely explored experimentally. In particular, the essential issue regarding the interplay between the Mott phase and dimensionality is yet to be clarified.

In this work, we address all the above key issues by performing a comprehensive angle-resolved photoemission spectroscopy (ARPES) study on epitaxially grown monolayer 1T-TaSe$_2$ and 1T-NbSe$_2$, and demonstrate the robust CDW-Mott phase under external perturbations such as heating and electron doping.

## Results and discussion

**Characterization of TaSe$_2$.** First, we discuss the electronic structure of monolayer 1T-TaSe$_2$ whose monolayer nature was confirmed by our scanning tunneling microscopy (STM) measurement (Supplementary note 1). Figure 1c displays the 3D ARPES intensity plotted as a function of 2D wave vector ($k_x$ and $k_y$) and binding energy $E_B$ measured at $T = 40$ K. One can clearly recognize a nearly flat band at $E_B \sim 0.3$ eV and dispersive holelike bands topped at the Γ point, which are ascribed to the Ta 5$d$ and Se 4$p$ bands, respectively[18]. The topmost Ta 5$d$ band does not cross the Fermi level ($E_F$) and exhibits an insulating gap of ~0.3 eV below $E_F$ at the Γ point. This gap is not assigned to a band gap, a substrate-induced gap, or a conventional CDW gap (Supplementary note 2), but to a Mott–Hubbard gap. This Mott gap is associated with the enhancement of $U/W$ caused by the hybridization of backfolded bands and the resultant band narrowing due to the $\sqrt{13} \times \sqrt{13}$ commensurate CDW (Fig. 1b), as in bulk nonmagnetic 1T-TaS$_2$[12,13] and a surface layer of bulk 1T-TaSe$_2$[21,22] as can be suggested from the overall similarity of experimental band dispersion (Supplementary Fig. S2). The gap size below $E_F$, called here $\Delta_{\text{Mott}}$, roughly corresponds to a half of the full Mott-gap size $2\Delta_{\text{Mott}}$ because $E_F$ is nearly located at the midpoint between the lower Hubbard band (LHB) and the upper Hubbard band (UHB) as suggested from the comparison of ARPES and tunneling spectroscopy data[20,23,24] (Supplementary note 3). As shown in Fig. 1c, a signature of the CDW is clearly seen as an apparent hybridization-gap discontinuity[21,22] in the band dispersion at $k \sim 2/3$ ΓM (red dashed line, see Fig. 1c). It is important to note that we could selectively fabricate a pure 1T-TaSe$_2$ phase (and also 1T-NbSe$_2$ phase, discussed later) with ease by controlling the substrate temperature[18]. This enables observation of a clear hybridization-gap discontinuity in our data as compared with a recent study, where admixture from the 1H-TaSe$_2$ phase made it difficult to see the discontinuity[20]. The STM image in Fig. 1d obtained in a spatial region of $8 \times 8$ nm$^2$ on a monolayer TaSe$_2$ island signifies a clear periodic modulation associated with the formation of CDW containing the hexagonal lattice of star-of-David clusters. We have confirmed that this lattice has a periodicity of $\sqrt{13} \times \sqrt{13}R13.9°$ expected for the formation of star-of-David lattice, as well visible as superspots in the Fourier transform image shown in Fig. 1e, in agreement with the previous STM study of monolayer 1T-TaSe$_2$[20].

The Ta 4$f$ core-level spectroscopy (Fig. 1f) signifies that the Ta 4$f_{5/2}$ and 4$f_{7/2}$ spin-orbit satellite peaks split into two subpeaks, as is clearly visible in the energy-distribution curve (EDC) at $T = 40$ K. Since the additional splitting of Ta-4$f$ peak is attributed to the different electron density at each Ta site[25,26] and/or the change in the chemical bonding of Ta atoms due to the formation of the star-of-David clusters, the core-level spectrum is consistent with our STM data that support the formation of the star-of-David clusters. On elevating temperature, we found that the lower-binding-energy subpeak of both the Ta4$f_{5/2}$ and 4$f_{7/2}$ components is gradually weakened, but the shoulder feature still remains even at $T = 400$ K. This implies that the Mott phase survives much above the room temperature. We will come back to this point later.

**Temperature dependence of the Mott gap.** The formation of CDW is further corroborated by the appearance of a LHB in the ARPES intensity at $T = 300$ K (Fig. 2b), similarly to the case at $T = 40$ K (Fig. 2a), because the Mott gap cannot be formed

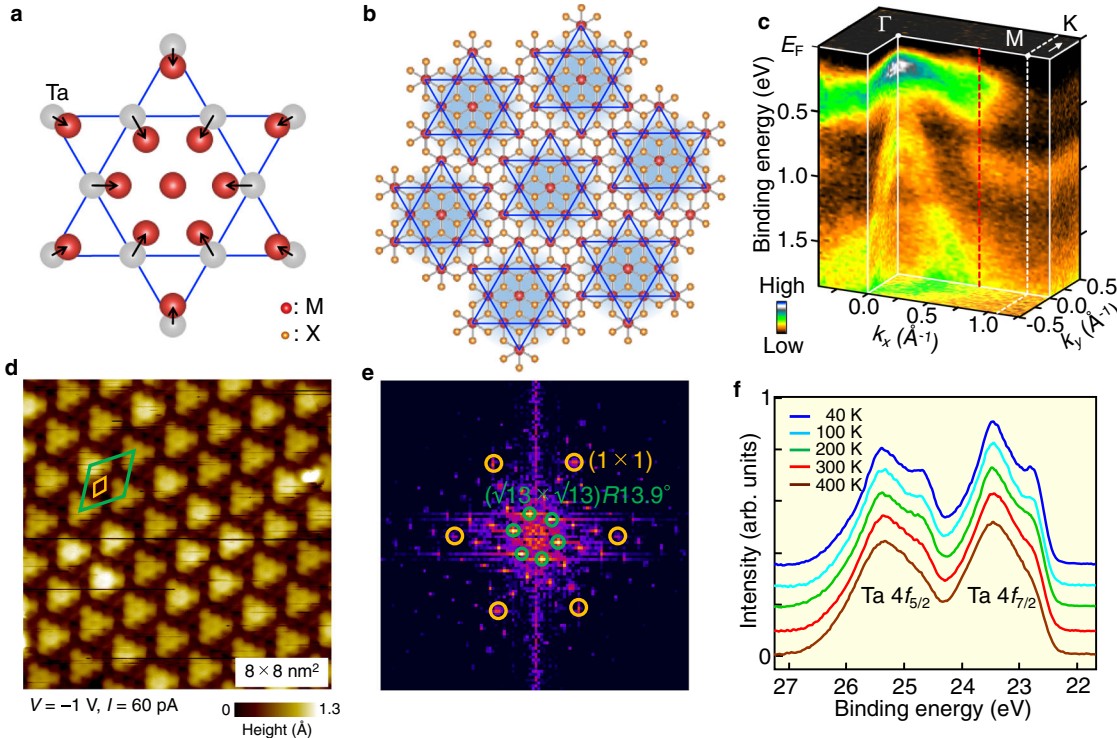

**Fig. 1 Schematics of star-of-David clusters and core-level photoemission spectrum of monolayer 1T-TaSe$_2$. a** Schematics of the displacement of Ta atoms in the star-of-David cluster. M and X represent transition-metal and chalcogen atoms, respectively. **b** Schematics of crystal structure for monolayer 1T-TaSe$_2$ and star-of-David clusters with the $\sqrt{13} \times \sqrt{13}$ periodicity. **c** 3D ARPES-intensity plot as a function of 2D wave vector ($k_x$ and $k_y$) and $E_B$ for monolayer 1T-TaSe$_2$ measured at $T = 40$ K with the He-I$\alpha$ line ($h\nu = 21.218$ eV). Hybridization gap ($k_x \sim 2/3$ $\Gamma$M) is indicated by red dashed line. **d** STM image in a surface area of $8 \times 8$ nm$^2$ for monolayer 1T-TaSe$_2$ on bilayer graphene measured at $T = 4.8$ K. **e** Fourier transform image of **d**. **f** Temperature dependence of EDC around the Ta-4$f$ core level measured with $h\nu = 260$ eV for monolayer 1T-TaSe$_2$.

without the CDW[12,13]. Intriguingly, the LHB survives even upto $T = 450$ K (the highest temperature in our experimental setup, see Fig. 2c), whereas the overall spectral feature becomes less clear. Such spectral feature at $T = 450$ K cannot be explained in terms of the absence of Mott gap and the persistence of CDW gap because of the following reason. In bulk TaSe$_2$[27], the LHB essentially vanishes at room temperature and a large metallic spectral weight emerges at $E_F$, in contrast to the low-temperature (70–220 K) data that display a peak associated with the LHB. Our ARPES data for monolayer 1T-TaSe$_2$ at room temperature resemble that of bulk TaSe$_2$ at low temperature (Fig. 2d), suggestive of the persistence of a Mott gap at $T = 450$ K (Supplementary note 4). The robustness of Mott gap is also seen from the detailed temperature dependence of EDC at the $\Gamma$ point in Fig. 2d. This is in stark contrast to bulk 1T-TaS$_2$ where a metallic Fermi edge is recovered at $T = 300$ K. Also, this is distinct from bulk 1T-TaSe$_2$ that shows a clear Fermi-edge cutoff even at $T = 30$ K (Fig. 2d) and hence, we compared the $T$-dependent behavior of the clear gap observed in monolayer 1T-TaSe$_2$ with bulk 1T-TaS$_2$.

To discuss more quantitatively the gap behavior, we plot the binding energy of the leading-edge midpoint of the EDC, $\Delta_{LEM}$, as a function of temperature for monolayer 1T-TaSe$_2$, and compare it with that for bulk 1T-TaS$_2$. We expect $\Delta_{LEM}$ to be directly related with the transport gap in the monolayer instead of the spectroscopic gap $\Delta_{Mott}$ (for the difference between $\Delta_{LEM}$ and $\Delta_{Mott}$, see Supplementary note 5). As one can see from the $\Delta_{LEM}^2$ vs $T$ plot in Fig. 2e, $\Delta_{LEM}^2$ for monolayer 1T-TaSe$_2$ shows a nearly linear behavior as a function of $T$ near $T_{CDW-Mott}$, and exhibits a finite value even at 450 K. This nearly linear behavior is also seen in bulk 1T-TaS$_2$ as shown in Fig. 2e, and was also reported for

bulk and monolayer 1T-TiSe$_2$ recently[28]. The $T$ dependence of $\Delta_{LEM}$ is well reproduced by a semiphenomenological BCS gap equation based on the mean-field approximation (blue solid curve) expressed as $\Delta_{LEM}(T)^2 - \Delta_{LEM}$ $(T_{CDW-Mott})^2 \propto \tanh^2$ $(A\sqrt{((T_{CDW-Mott}/T)-1)})$, where $\Delta_{LEM}$, $T_{CDW-Mott}$, and A are the binding energy of the leading-edge midpoint, the CDW–Mott transition temperature, and the proportional constant, respectively[28], which was recently used to characterize bulk and monolayer 1T-TiSe$_2$ (note that the observed temperature dependence of EDC can hardly be explained with the thermal broadening; for details, see Supplementary note 4). From the numerical fittings, the transition temperature was estimated to be $T_{CDW-Mott} \sim 530$ K for monolayer 1T-TaSe$_2$, and this is much higher than that obtained for bulk 1T-TaS$_2$ (<200 K; red circles and curves). The present results suggest that the CDW-Mott-transition temperature $T_{CDW-Mott}$ of monolayer TaSe$_2$ is very high, being drastically higher than that of bulk TaS$_2$ ($T_{CDW-Mott} \sim 200$ K)[12] and a surface layer of bulk TaSe$_2$ ($T_{CDW-Mott} \sim 260$ K)[27,29] (note that a consensus has not been reached for the exact $T_{CDW-Mott}$ value at the surface of TaSe$_2$; Supplementary note 6 and Fig. S5). In contrast, the increase in $T_{CDW}$ of 1T-TiSe$_2$ in going from bulk ($T_{CDW} \sim 200$ K) to monolayer ($T_{CDW} \sim 235$ K) was small[28,30]. If we consider $\Delta_{LEM}$ to be a measure of half the transport gap (since $E_F$ lies in the middle of the gap, as discussed earlier), $2\Delta_{LEM}(T=0)/k_B T_{CDW-Mott} \sim 12$, which is significantly larger than the weak-coupling result of $\sim3.52$. Thus, monolayer 1T-TaSe$_2$ can be classified as a strongly coupled CDW–Mott phase. It is noted that our time-resolved ARPES experiment suggests that the LHB of monolayer TaSe$_2$ survived after photoexcitation even when we adopted the maximum pump fluence above which monolayer samples were damaged

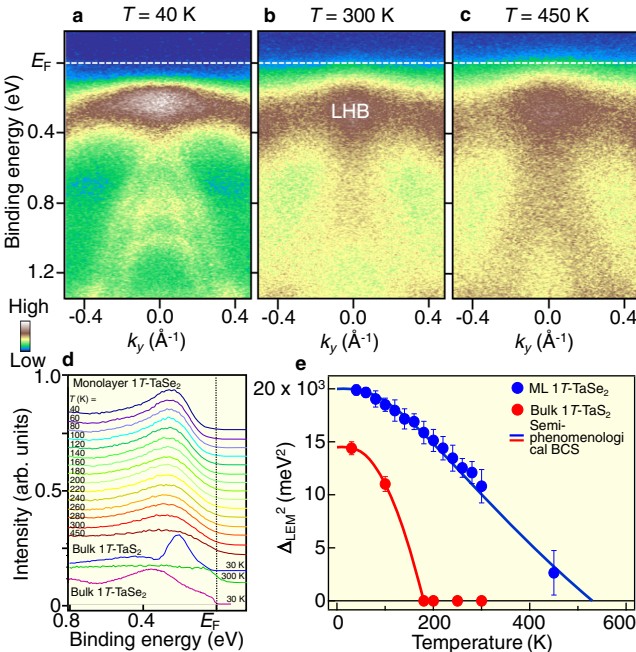

**Fig. 2 CDW–Mott phase of monolayer 1T-TaSe₂ robust against temperature variation. a-c** Near-$E_F$ ARPES intensity along the ΓK cut measured at $T = 40$, 300, and 450 K, respectively. **d** Temperature dependence of EDC at the Γ point. EDC for bulk 1T-TaS₂ ($T = 30$ and 300 K) and bulk 1T-TaSe₂ ($T = 30$ K) is also shown as a reference. **e** Squared leading-edge midpoint $\Delta_{LEM}$ at the Γ point plotted against $T$ for monolayer 1T-TaSe₂ (blue circles), together with the numerical fitting results with the semiphenomenological (blue solid curve) BCS gap functions. $\Delta_{LEM}$ and fitting results are also plotted for bulk 1T-TaS₂ (red). We have obtained {$T_{CDW-Mott}$, $A$} = {530 K, 1.01} and {180 K, 1.50} for monolayer 1T-TaSe₂ and bulk 1T-TaS₂, respectively. Error bars in **e** reflect the uncertainties originating from the energy resolution and the standard deviation in the peak positions of EDCs.

(this maximum pump fluence is lower than that in bulk TaS₂;[31-36] for details, see Supplementary note 7 and Fig. S6).

**Carrier doping effect to the Mott gap.** Next, we show the robustness of the LHB against carrier doping. Figures 3a–c show the evolution of ARPES intensity as a function of potassium (K) coverage $d_K$ [0 (pristine), 3.2 and 6.4 × 10¹³ atoms/cm²] and the corresponding second-derivative intensity plots (d–f) (for details of the $d_K$ estimation, see Methods section). Upon K deposition of $d_K = 3.2 × 10^{13}$ atoms/cm², which corresponds to ~50% of the star-of-David density (Fig. 3b), the band structure is shifted downward as a whole with respect to pristine TaSe₂ (Fig. 3a) due to the electron doping from K atoms. This suggests that the K deposition dopes electron carriers into a whole monolayer film, as in the case of other monolayer TMD films such as TiSe₂ where K deposition causes an overall downward shift of the band structure and disappearance of CDW[37]. On further depositing K atoms (Fig. 3c), the spectral feature becomes significantly broad due to the strong electron scattering by the K-induced disorder, whereas a broad hump originating from the LHB is still seen at $E_B$ ~ 0.6 eV in the EDC in Fig. 3g. The LHB and its systematic downward shift are better visualized in the second-derivative-intensity plots in Fig. 3d–f. A careful look at the intensity for $d_K = 6.4 × 10^{13}$ atoms/cm² (Fig. 3f) reveals a bright intensity in the vicinity of $E_F$, which originates from a finite Fermi-edge cutoff, as also seen in the EDC (green curve) in Fig. 3g.

To discuss the spectral evolution upon K dosing in more detail, we have analyzed the spectral weight at $E_F$ relative to that of LHB, $I_{EF}/I_{LHB}$, as a function of K coverage $d_K$. As shown in Fig. 3h, the $I_{EF}/I_{LHB}$ value (red circles) does not exhibit a monotonic behavior as a function of $d_K$, showing a minimum at $d_K = 3.2 × 10^{13}$ atoms/cm². The nonzero value for $d_K = 0$ atoms/cm² may be associated with the tail of LHB extending toward $E_F$, as also seen in the EDC (Fig. 3g). We found that the $I_{EF}/I_{LHB}$ value in monolayer for both $d_K = 3.2$ and $6.4 × 10^{13}$ atoms/cm² is larger than that of the CDW–Mott insulator phase ($T = 30$ K) in bulk TaS₂ (blue square), implying a possible metallic behavior. We have confirmed that such a difference between monolayer and bulk is not associated with the difference in the experimental conditions. This can be seen from Fig. 3h where $I_{EF}/I_{LHB}$ at $T = 300$ K for monolayer 1T-TaSe₂ ($d_K = 0$) obtained with He lamp (green circle) and synchrotron radiation (red circle) well coincide with each other within our experimental uncertainty.

A simple explanation to account for the observed metallic component may be electron occupation of the UHB. Since the energy position of LHB shifts from 0.28 eV to 0.75 eV with K deposition (Fig. 3g), one would expect the UHB to appear below $E_F$ in the $d_K = 6.4 × 10^{13}$ atoms/cm² sample since the full Mott gap is estimated to be 0.5 eV (Supplementary note 3). However, we found no evidence for the prominent peak from the UHB. This is reasonable because the high Coulomb cost $U$ to populate an electron to the UHB does not guarantee the rigid-band-like electron doping. As an alternative possibility, a metallic K component due to the high density of K atoms (which produces an angle-integrated-type background with a weak Fermi-edge cutoff in EDC) or in-gap states (with suppressed quasiparticle intensity and reduced lifetime) could be conceived. The latter possibility is expected from the Hubbard model for the doped Mott insulator and was observed in spectroscopic studies of cuprates[38].

As shown in Fig. 3h, the $I_{EF}/I_{LHB}$ value in the monolayer sample of $d_K = 6.4 × 10^{13}$ atoms/cm² is much smaller than that in the bulk counterpart with the fully melted CDW–Mott state at $T = 300$ K (yellow square). This implies that the framework of LHB itself is still maintained in the monolayer even when the system likely becomes metallic upon electron doping, in contrast to the bulk counterpart where even a small amount of electron doping breaks the LHB and leads to the occurrence of superconductivity[39-42]. In particular, doping electrons by substitution of magnetic Fe ions in 1T-Fe$_x$Ta$_{1-x}$S₂ was shown to result in a dispersive electron band accompanied by a destruction of the LHB even with 1% Fe substitution[40,41]. A plausible mechanism of such fragileness in bulk is associated with the Fermi-surface nesting condition that is sensitive to the carrier doping, and has been discussed as a primary cause of incommensurate CDW and the resultant Mott phase in bulk 1T-TaSe₂[43,44]. In this regard, the monolayer data are puzzling and surprising because the LHB still survives even when the nesting condition is modified by the electron doping. In any case, the survival of LHB suggests that the electron correlation is still strong even in the doped monolayer sample.

A recent generalized-gradient approximation band-structure calculation with on-site Coulomb interaction (GGA + U) for isoelectronic 1T-NbSe₂[45] has reproduced the LHB and Mott gap, consistent with the ARPES data. It is noted though, while the GGA + U study and a very recent DFT (density-function-theory) calculation with GGA on monolayer 1T-TaSe₂[46] suggested a spin-1/2 ferromagnetic Mott-insulator phase, our experimental attempt to detect possible ferromagnetism was not successful [since the detection of ferromagnetism by macroscopic magnetization measurement is difficult for monolayer samples, we carried out a very primitive experiment by just putting a strong

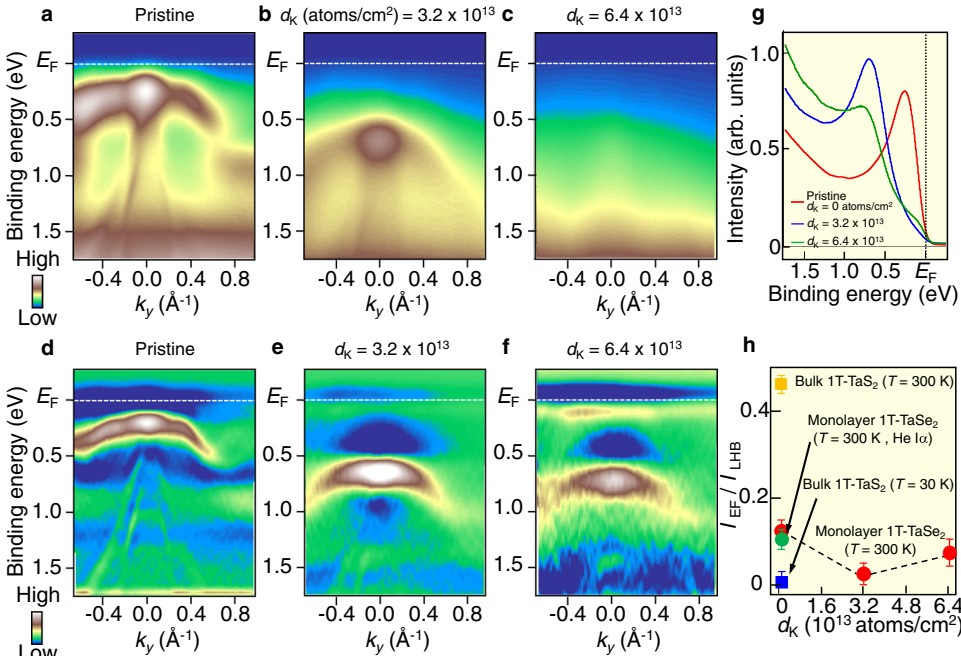

**Fig. 3 CDW–Mott insulator phase in monolayer 1T-TaSe$_2$ robust against electron doping. a–c** K-deposition dependence of ARPES intensity along the ΓK cut for monolayer 1T-TaSe$_2$ [potassium coverage $d_K = 0$ (pristine), $3.2 \times 10^{13}$, and $6.4 \times 10^{13}$ atoms/cm$^2$, respectively], measured at $T = 300$ K with $h\nu = 51$ eV. **d–f** Same as **a–c**, but obtained by taking the second derivative of EDCs. **g** EDCs at the Γ point for each $d_K$. **h** Plots of intensity at $E_F$ with respect to that at LHB, $I_{EF}/I_{LHB}$, as a function of $d_K$, estimated from the EDCs in **g**. The values for bulk TaS$_2$ measured at $T = 30$ and 300 K are also plotted. Error bars reflect the uncertainties originating from the energy resolution and statistics of data. Asymmetry in the intensity profile in **a** and **b** is associated with the inequivalent photoelectron matrix-element effect between positive and negative $k_y$'s (Supplementary note 8 and Fig. S7).

Nd magnet (magnetic field ~ 500 mT) on top of a film to detect possible attractive force]. A good agreement of the overall band structure in the Mott-insulator phase between monolayer TaSe$_2$ and bulk nonmagnetic 1T-TaS$_2$[41] may support the nonmagnetic ground state of monolayer TaSe$_2$, although this point needs to be verified in future, e.g., by X-ray magnetic circular dichroism measurement.

**Comparison between TaSe$_2$ and NbSe$_2$.** Now that the survival of LHB in various conditions is established for monolayer TaSe$_2$, next, we explore the CDW–Mott phase in a cousin material, monolayer 1T-NbSe$_2$. One can immediately recognize in the ARPES intensity along the ΓK cut in Fig. 4a that the LHB is well seen at $T = 40$ K in monolayer NbSe$_2$. The LHB survives even at $T = 450$ K (Fig. 4b), as is also visible in the EDC at the Γ point in Fig. 4c. A side-by-side comparison of the valence-band ARPES intensity along the ΓM cut between monolayer TaSe$_2$ and NbSe$_2$ in Fig. 4d, e reveals several common features, such as a nearly flat LHB, dispersive hole-like Se-4$p$ bands, and a discontinuity of band dispersion at $k_x \sim 2/3$ ΓM caused by the hybridization-gap opening due to the CDW. These results demonstrate that the robust Mott-insulator phase coexisting with CDW upon heating is a common characteristic for monolayer TaSe$_2$ and NbSe$_2$.

In the following, we discuss why the LHB in monolayer TaSe$_2$ and NbSe$_2$ is robust unlike in the bulk counterpart. Since one of the key parameters to trigger a CDW–Mott transition is the effective Coulomb interaction $U/W$, it is important to discuss the independent roles of how $U$ and $W$ are independently affected on going from the bulk 3D structure to the monolayer 2D case. The effective on-site Coulomb correlation energy can be described by the equation $U = E_I - E_A - E_{Pol}$ where, $E_I$ is the ionization energy, $E_A$ is the electron affinity, and $E_{Pol}$ is the polarization energy that arises from screening due to any electronic perturbation such as removing or adding an electron. This screening causes a strong

reduction of $U$ compared with the bare Coulomb interaction $U_{bare} (= E_I - E_A)$. For example, based on a one-band Hubbard model, $U$ for the Cu site gets reduced to ~4 eV in La$_2$CuO$_4$ compared with $U_{bare}$ ~20 eV for Cu atom[47]. For 4$d$ and 5$d$ transition metals, $U$ is expected to be still lower. For 1T-TaS$_2$, 1T-TaSe$_2$, and 1T-NbSe$_2$, typical values of $U$ reported in the literature range from ~0.4/0.7 eV (in the GW approximation/DFT-DMFT approximation[16,48]) to ~2.0/2.8 eV (in GGA + $U$)[20,49,50]. Considering the role of screening in monolayer compared to the bulk case, while the intralayer $E_{Pol}$ is expected to show negligible changes in the monolayer case, the interlayer $E_{Pol}$ would be suppressed as there are no other layers and the interaction with the substrate is weak, resulting in an effective increase in $U$ compared with the bulk.

Similarly, since there is no out-of-plane or interlayer hopping in the monolayer, i.e., the intrinsic bulk interlayer bandwidth $W_{out}$ is absent, the net effective bandwidth $W$ will get reduced. It was suggested from the first-principles band-structure calculations that, although the in-plane bandwidth $W_{in}$ becomes significantly small (~0.08 eV[51]) due to the band reconstruction associated with CDW, the out-of-plane bandwidth $W_{out}$ (~0.54 eV[51]) does not suffer from a strong band-narrowing effect because of the in-plane nature of CDW. In this case, the dominant channel to determine the total $W$ is the interlayer hopping (Fig. 5b) (note that the inter- and intralayer hopping channels do not contribute in an additive way to the bandwidth, but one can still discuss which plays a dominant role). It is thus inferred that the $U/W$ value in the bulk is largely governed by the interlayer hopping and the bulk is located on the verge of the CDW–Mott transition ($U/W \sim 1.3$;[48] note that $U$ is ~0.7 eV in TaS$_2$[16]). On the other hand, in monolayer, the interlayer hopping is intrinsically absent (Fig. 5b) and the net $W$ is simply associated with the intralayer hopping. Thus, both the increase in $U$ and decrease in $W$ are expected to positively work together to

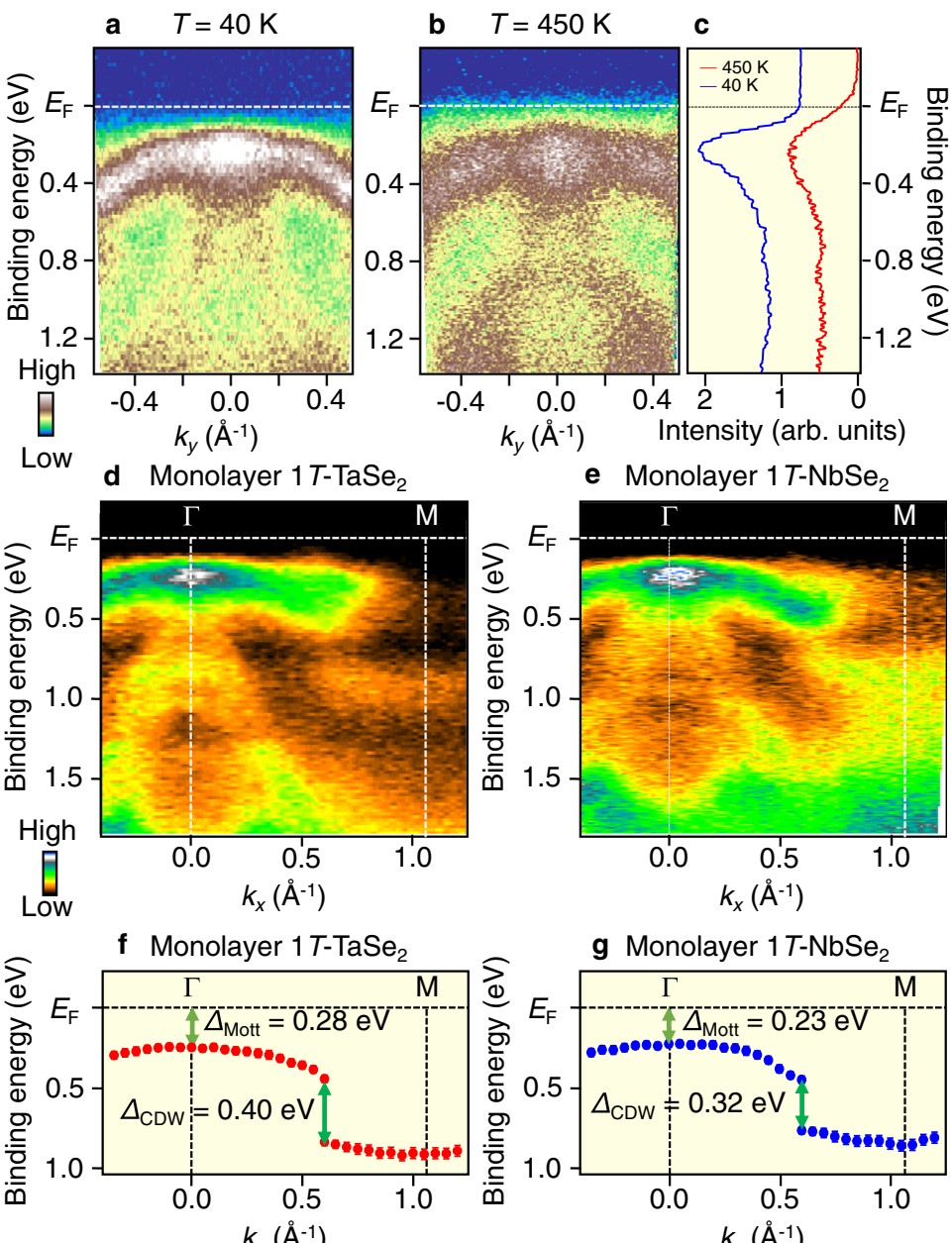

**Fig. 4 Comparison of the CDW–Mott phase between monolayer NbSe₂ and TaSe₂. a, b** ARPES intensity of monolayer 1T-NbSe₂ at $T = 40$ and 450 K, respectively, measured along the ΓK cut. **c** EDCs at the Γ point at $T = 40$ and 450 K. **d, e** Valence-band ARPES intensity along the ΓM cut for monolayer NbSe₂ and TaSe₂, respectively. **f, g** Experimental band dispersion extracted by tracing the peak position in EDCs for monolayer TaSe₂ and NbSe₂, respectively, highlighting the quantitative difference in the magnitude of the Mott–Hubbard gap ($\Delta_{\mathrm{Mott}}$) and the hybridization gap associated with CDW ($\Delta_{\mathrm{CDW}}$). Error bars in **e** reflect the uncertainties originating from the energy resolution and the standard deviation in the peak positions of EDCs.

efficiently increase $U/W$, leading to the robust CDW–Mott phase far above bulk $T_{\mathrm{Mott}}$. We remark here that it is difficult to experimentally determine $W$ by simply tracing the continuously dispersing bands in the experiment, because such bands are composed of multiple subbands reconstructed by the CDW and the intensity of bands is often suppressed in the region away from the original unfolded band[33]. The Mott transition is associated only with a half-filled subband crossing $E_F$, which has a narrow in-plane bandwidth $W_{\mathrm{in}}$[33], although previous ARPES studies above $T_{\mathrm{Mott}}$ on bulk TaS₂ (e.g.,[21,26,27,41]) were unable to resolve this band, probably because of the smearing of fine-band structure by, e.g., thermal broadening.

A comparison of characteristic energy scales between monolayer TaSe₂ and NbSe₂ reveals an intriguing aspect of the

CDW–Mott phase in the two systems (Fig. 4d and e). As shown in the ARPES-derived band dispersions in Fig. 4f and g, a half of the full Mott gap, $\Delta_{\mathrm{Mott}}$, in TaSe₂ ($0.28 \pm 0.02$ eV) is slightly larger than that in NbSe₂ ($0.23 \pm 0.02$ eV). Also, the hybridization gap $\Delta_{\mathrm{CDW}}$ in TaSe₂ ($0.40 \pm 0.03$ eV) is larger than that in NbSe₂ ($0.32 \pm 0.03$ eV). According to the general trend of $U$ in $3d$–$4d$–$5d$ electron systems, $U$ for the Nb-$4d$ orbital is expected to be larger than that for the Ta-$5d$ orbital. Band-structure calculations suggested that the in-plane bandwidth of Nb $4d$ band in the normal state of monolayer NbSe₂ ($\sim$2.2 eV)[49] is smaller than that in monolayer TaSe₂ ($\sim$2.7 eV)[52]. It is expected from a simple band-folding picture that the bandwidth in the CDW phase $W_{\mathrm{in}}$ is also smaller in NbSe₂. Although this argument suggests a larger $U/W$ and a resultant more stable CDW–Mott phase in NbSe₂

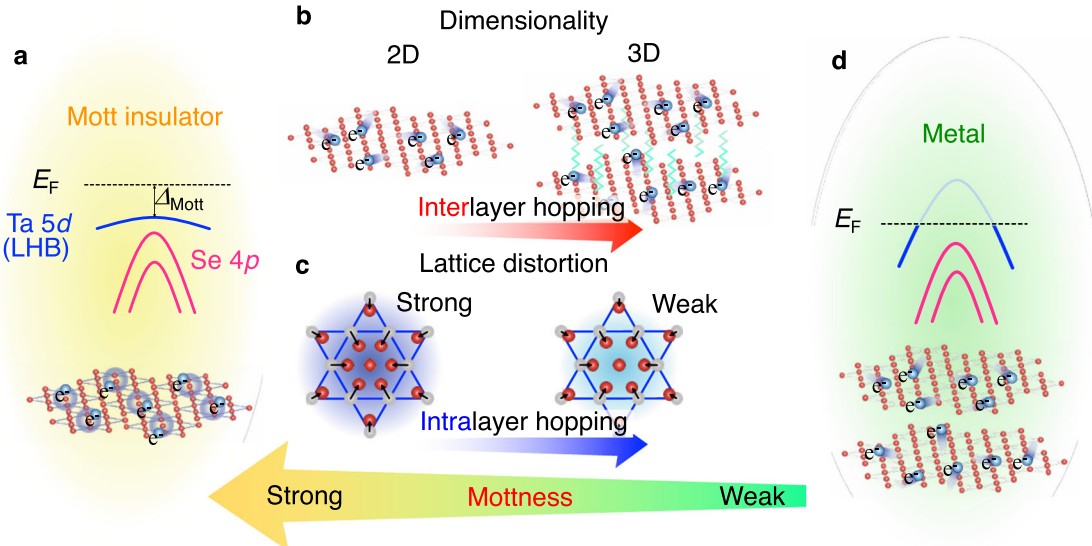

**Fig. 5 Realization of 2D CDW–Mott insulator phase assisted by strong lattice distortion. a** Schematics of (top) the band dispersion and (bottom) the star-of-David clusters in the robust CDW–Mott phase in monolayer TaSe$_2$. **b** Illustration to compare the interlayer hopping in the crystal in 2D and 3D systems. **c** Illustration of strong vs weak lattice distortion in the star-of-David cluster and its relationship with the intralayer hopping in the CDW phase. **d** Schematics of (top) the normal-state band dispersion and (bottom) the bulk crystal structure without CDW formation.

than in TaSe$_2$, the observed smaller $\Delta_{Mott}$ (0.23 eV) in NbSe$_2$ (this also implies lower $T_{Mott}$) apparently contradicts with the above simple argument. This discrepancy may be reconciled by taking into account the observed larger $\Delta_{CDW}$ (0.40 eV) in TaSe$_2$, which suggests a smaller intralayer hopping and as a result a larger $U/W$ compared with NbSe$_2$ (note that while the bandwidth of LHB is estimated to be 0.19 and 0.21 eV for TaSe$_2$ and NbSe$_2$, respectively, these values cannot be regarded as $W$ and one needs to know the original bare $W$ without any influence from the Mott gap). This suggests that the lattice displacement in the star-of-David cluster is stronger in monolayer TaSe$_2$ (Fig. 5c; left), which is also inferred from the stronger metallic-bonding character of Ta than that of Nb. The stronger periodic lattice distortion due to the CDW in TaSe$_2$ is also supported by the observation of more pronounced hybridization-gap discontinuity around $k \sim 2/3$ ΓM in TaSe$_2$, as seen from the stronger intensity suppression at $E_B \sim 0.6$ eV in Fig. 4d than that in Fig. 4e (note that some other experiments must be carried out to firmly establish the stronger CDW in TaSe$_2$). It is also noted that $\Delta_{CDW}$ of monolayer TaSe$_2$ (0.40 eV) is slightly larger than that of bulk TaSe$_2$ (0.37 eV);[22] this suggests a stronger lattice distortion in monolayer systems, consistent with GGA + U calculations discussed above[45]. All these arguments suggest that the robust CDW–Mott insulator phase of monolayer TaSe$_2$ and NbSe$_2$ (Fig. 5a) is caused by the disappearance of interlayer hopping assisted by a strong lattice distortion. In other words, the robust CDW in monolayer TaSe$_2$ and NbSe$_2$ is derived from a combination of electron correlations, a strong lattice distortion, and the absence of interlayer hopping. It is emphasized that such properties are all linked to the electron hopping (or electron kinetic energy) of the system (Fig. 5b and c), and the controllability of the Mottness (i.e., strength of the Mott phase) lies on how to effectively manipulate both inter- and intralayer hopping (Fig. 5a–d).

The present study has established an effective means to stabilize the CDW–Mott phase in terms of band engineering. Also, the discovery of the robust CDW–Mott phase far above the room temperature is useful for developing practical CDW–Mott insulator-based ultrathin nanoelectronic devices. It would be very interesting to explore the superconductivity in a metallic state near the Mott phase.

## Methods

**Sample preparation**. Monolayer 1T-TaSe$_2$ and 1T-NbSe$_2$ films were grown on bilayer graphene/6H-SiC by using molecular-beam-epitaxy (MBE) method in an ultrahigh vacuum (UHV) of $3 \times 10^{-10}$ Torr. As for the monolayer NbSe$_2$ film, we have adopted the same procedure to grow monolayer 1T-NbSe$_2$ established in our previous studies where the 1T structure, $\sqrt{13} \times \sqrt{13}$ lattice reconstruction, and its monolayer nature were already clarified[19,53]. To fabricate a monolayer TaSe$_2$ film, we have also followed the fabrication method established by ourselves[18]. Specifically, bilayer graphene was prepared by annealing an $n$-type Si-rich 6H-SiC(0001) single-crystal wafer, with resistive heating at 1100 °C in ultrahigh vacuum better than $1 \times 10^{-10}$ Torr for 30 min. A monolayer TaSe$_2$ (NbSe$_2$) film was grown by evaporating Ta (Nb) on the bilayer graphene substrate kept at 560 °C (580 °C) under a Se atmosphere[18,19]. The as-grown film was subsequently annealed at 400 °C for 30 min. The growth process was monitored by reflection high-energy reflection diffraction (RHEED). The film thickness was estimated by a quartz-oscillator thickness meter, scanning tunneling microscopy (STM), and atomic force microscopy (AFM). Based on our experience of fabricating various monolayer TMD films such as NbSe$_2$, VTe$_2$, VSe$_2$, and TiSe$_2$[18,19,54,55], a monolayer film is formed immediately after the disappearance of buffer-layer-originated $6\sqrt{3} \times 6\sqrt{3}$ RHEED pattern upon coevaporation of transition-metal and chalcogen atoms. We have judged the thickness of 1T-TaSe$_2$ and NbSe$_2$ films by monitoring this disappearance in the RHEED pattern. After the fabrication by the MBE method, the films were transferred to the ARPES-measurement chamber without breaking the vacuum. We have calibrated the deposition rate of K atoms by calculating the volume of $\pi$-band Fermi surface at the K point in bilayer graphene on SiC with keeping the same evaporation rate as that in the case of monolayer TaSe$_2$, and it is estimated to be $1.6 \times 10^{13}$ atoms/cm$^2$/min. We have deposited K atoms for 2 and 4 min. that corresponds to the K coverage $d_K$ of 3.2 and $6.4 \times 10^{13}$ atoms/cm$^2$, i.e., ~50 and ~100% of the star-of-David density, respectively. Thus, the amount of K dosing with respect to the star-of-David density is sufficient to achieve a sizable electron doping into monolayer 1T-TaSe$_2$.

**ARPES and STM measurements**. ARPES measurements were carried out using an MBS-A1 electron-energy analyzer with a high-flux helium-discharge lamp and a toroidal grating monochromator at Tohoku University and an Omicron-Scienta R4000 electron-energy analyzer with synchrotron radiation at Taiwan Light Source (TLS), National Synchrotron Radiation Research Center (NSRRC). The energy and angular resolutions were set to be 12.5–40 meV and 0.2°, respectively. Core-level photoemission-spectroscopy measurement was performed at BL28A with micro-focused beam spot in Photon Factory. Time-resolved ARPES measurements were carried out at Tsinghua University using an Omicron-Scienta DA30-L-8000 electron-energy analyzer and a Ti:sapphire oscillator that produces femtosecond pulses from 700 to 980 nm at 76 MHz repetition rate with pulse duration of ~130 fs. The time resolution was 480 fs when the probe photon energy was set to 6.2 eV[56]. The infrared laser was frequency-quadrupled using BBO and KBBF crystals to produce an ultraviolet probe light from 177.5 to 230 nm. The beam sizes of pump and probe beam were set to ~45 μm and ~15 μm (full width at half maximum), respectively. The wavelengths of pump and probe beam were set to 800

and 200 nm, respectively. The repetition rate was set to 1000 kHz using a pulse picker. The energy and angular resolutions were set to be 8 meV and 0.1°, respectively. The Fermi level ($E_F$) of the sample was calibrated with a gold film deposited onto the substrate. To avoid contamination of the sample surface in ex situ ARPES measurements, we covered the film with amorphous Se immediately after the epitaxy, transferred it to a separate ARPES chamber, and then decapped the amorphous Se film by heating under UHV.

STM measurements were carried out using a custom-made ultrahigh vacuum STM system[57]. Se capping layers for surface protection of TaSe$_2$ films were removed in the STM chamber by Ar$^+$ ion sputtering for 30 min and annealing at 200 °C for 60 min. STM measurements were performed with PtIr tips at 4.8 K under UHV below $2.0 \times 10^{-10}$ Torr. All STM images were obtained in constant-current mode.

## Data availability
The data that support the findings of this study are available from the corresponding author upon reasonable request.

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

## Acknowledgements

We thank Takumi Sato, T. Taguchi, and C.-W. Chuang for their assistance in the ARPES measurements. We also thank NSRRC-TLS for access to beamline BL21B1. This work was supported by JST-CREST (no. JPMJCR18T1), JST-PRESTO (no. JPMJPR20A8), Grant-in-Aid for Scientific Research on Innovative Areas "Topological Materials Science" (JSPS KAKENHI Grant numbers JP15H05853 and JP15K21717), Grant-in-Aid for Scientific Research (JSPS KAKENHI Grant numbers JP21H04435, JP17H01139), National Natural Science Foundation of China (11725418, 11427903), Ministry of Science and Technology of China (2016YFA0301004, 2015CB921001), Beijing Advanced Innovation Center for Future Chip (ICFC), Tsinghua University Initiative Scientific Research Program, Tohoku-Tsinghua Collaborative Research Fund, Grant for Basic Science Research Projects from the Sumitomo Foundation, Research Foundation of the Electrotechnology of Chubu, Ministry of Science and Technology of the Republic of China, Taiwan, under contract no. MOST 108-2112-M-213-001-MY3, and World Premier International Research Center, Advanced Institute for Materials Research. Y. N. and T. K. acknowledge support from GP-Spin at Tohoku University. A.C. and C.M.C. thank the Ministry of Science and Technology (MOST) of Taiwan, Republic of China, for financially supporting this research under Contract No. MOST 109-2911-I-213-501.

## Author contributions

The work was planned and proceeded by discussion among Y.N, K.S. and T.S. Y.N. and K.S. fabricated ultrathin films. Y.N., K.S., A.C., C.B., S.Z., P.C., C.C., T.K., Y.S. and S.Z. performed the ARPES measurements. H.O. and T.F. performed the STM measurements. Y.N., A.C., T.T. and T.S. finalized the paper with inputs from all the authors.

## Competing interests

The authors declare no competing interests.
