## [Peer Review File · Nature Communications]

REVIEWER COMMENTS

Reviewer #1 (Remarks to the Author):

The manuscript of Nakata et al. discuss the robustness of charge-density-wave/Mott phases in monolayer (ML) transition metal dichalcogenides (TMDs). The topic is of large interest to a wide scientific community and the methods are appropriate. There are, however, several points that require additional clarifications before granting publication to Nat. Comm.

- Samples -

Concerning the ML specimens: have the authors performed any LEED / STM / AFM analysis on "these" specific samples (similar to what is reported in Ref. [17-18]) in order to ensure the formation of 1T phase and the $\sqrt{13 \times 13}$ lattice reconstruction? (in particular, NbSe₂ is very likely to grow as 2H). Furthermore, is there any evidence that TaSe₂ and NbSe₂ are truly ML? A bilayer system would definitely change their conclusions about the intra-layer vs inter-layer hopping. I believe some stronger proofs should be given other than the ARPES comparison reported in SI note 5.

- Time-resolved ARPES -

At line 150 the authors claim a 5.2 work function for TaSe₂ and thus a binding energy range of 1 eV (given the 6.2 eV photons), but Fig. 2f-2g are not in agreement with such statement.

At line 164-165 they also claim that CDW-Mott phase is hardly melted, in contrast to bulk TaS₂. However, this fact depends critically on the pump fluence (they declare 0.26 mJ/cm² in Fig.2) and experimental conditions. There are several experimental evidences (see f.i. recent Dragan Mihailovic's works) showing that to trigger (electronic) phase transitions in Ta-based TMDs a pump fluence of the order of 1 mJ/cm² is required. Thus, it is not surprising that the authors do not see any "melting" of CDW-Mott phase. In order to demonstrate robustness against photo-excitation a more extensive investigation is necessary. Besides, in SI note 6, the authors report an estimate of the absorbed energy/pulse based on the optical properties of bulk TaS₂. Although optical data and formulas should be specified, a ML film might behave very differently from a bulk and the substrate underneath might play a relevant role in the optical response. Finally, the fluence comparison with Ref.[32] is not the most appropriate: see f.i. recent works of Kapteyn-Murnane group (Sci. Adv. 5, eaav4449 (2019), PNAS 117 8788 (2020)) dealing with TaSe₂.

Considering that (i) ARPES with 6.2 eV photons (Fig. 3f) does not bring substantial additional information compared to He lamp ARPES and that (ii) the time-resolved experiment is not truly

necessary for the goal of this work, I suggest to remove this part and maybe address the time-resolved features in a different publication.

- Potassium doping -

At line 172-173 the potassium dose is given in terms of "deposition time" which is not particularly useful (it depends on the experimental conditions) and very qualitative. The relevant information is the K coverage in ML. The authors should at least provide an estimate of the dose to allow a quantitative analysis of the electronic doping. As an example: is the K coverage (atoms/cm²) comparable to the "Stars of David" density (roughly 7×10^{13} cm⁻² in TaSe₂)?

- Hubbard bands -

Referring to the discussion starting at line 178 (and related Fig.3d-f), the LHB is about 250 meV below E_F in the pristine sample. This means that the UHB should be roughly 250meV above E_F (as properly stated in SI note 2). After 4 min K deposition, the LHB shifts to 750 meV below E_F . Therefore, the UHB should be well below E_F . Authors claim that the spectral weight appearing close to E_F in Fig. 3f might be the tail of the UHB (line 191). However, I strongly doubt this could be possible: the fully populated LHB is actually a half-filled band (1 electron per site). Any hopping event would lead to double occupancy (e-e at one site and h-h at a neighboring site). This excited state has a Coulomb cost (U) that ends up in populating the UHB. I cannot imagine a strongly or even fully populated UHB below E_F . Instead, electron doping (due to K deposition) would most likely turn the (half-filled) Hubbard band into a standard (fully occupied) metallic band, which is energetically more convenient. The spectral weight at E_F could be simply due to potassium (as more reasonably stated in line 192).

- Doping comparison -

See lines 202-204: there is a crucial difference between electron doping via alkali deposition and electron doping in a bulk, such as reported in Ref.[37-38]. In the first case the doping is surface-confined, it leads to space charge layer, band bending and so forth. In the second case the doping is intrinsically induced during the growth process and has no depth dependence (no space charge layer, no band bending). So, these are different physical conditions leading to different physical properties. In particular, the potassium-induced shift of the binding energy seen with ARPES (Fig. 3 d-f) points at the formation of a space charge layer with a downward band bending (as expected for an electron donor doping such as K, see f.i. the work on black phosphorus by Kim et al. Science 349,

723 (2015)). This fact casts some doubts on the true ML nature of the TMD film and on the interlayer hopping argument (see my first comment about the samples). Authors should clarify this point.

- Additional comments –

line 28: "...up to high temperature.". A more specific statement is required.

line 51: "...U/W exceed the critical value". A Reference is required.

lines 114-117: "Such spectral features..." The absence of Mott gap and the persistence of CDW phase was seen f.i. in Ref.[27] at room temperature. Authors should clarify.

lines 125-126: "We expect...": what do authors mean by "transport gap" and "spectroscopic gap minimum Δ_{Mott} "? Data in Fig.2e seem to follow a BCS-like behavior, which normally is attributed to the CDW gap (in fact the model in Ref.[26] refers to CDW), but these data refer to the leading edge at Gamma that is the LHB. This is confusing. Authors should clarify.

lines 129-130: "This is also true..." what is also true? the linear behavior (I guess)? or the finite value at high temperature? Sentence is misleading.

line 132: authors should report the equations.

line 198 and Fig. 3f: I find this comparison misleading and too heterogeneous to be significant. There are data extrapolated from different experiments, different authors on different samples. I strongly suggest to reconsider the use of this Figure.

line 219: "... possible ferromagnetism was not successful". Authors should specify what kind of attempt they made.

line 237: "...between monolayer and bulk." A Reference is required.

line 257: "...previous ARPES studies above T_{Mott} ". A Reference is required.

line 273: "... a smaller interlayer hopping...". This is not clear: authors are arguing on the Mott and CDW gaps of the ML systems and infer about the magnitude of the interlayer hopping (which in a ML cannot exist!)

line 278: "The stronger CDW...": in what sense the CDW is stronger? Do authors refer to periodic lattice distortion?

line 320: What is the time resolution (or UV pulse duration) of the time-resolved ARPES setup?

Fig. 1d, 2d, 2i, 3g, 4c: y-axis should report numbers (even if a.u. are used).

Fig.2e, 3h, 4f-g: all experimental dots should have corresponding error bars.

Reviewer #2 (Remarks to the Author):

Sato et al. present a detailed experimental study on 1T-TaSe_2 and 1T-NbSe_2 being prepared as monolayers epitaxially grown on bilayer graphene. The graphene itself was prepared on SiC(0001) single crystals.

In the photoemission data the authors observe characteristic spectroscopic differences compared to respective the bulk material, in particular in the temperature dependence of the features in the valence band. The observations are thoroughly discussed in the well established picture of for the non-magnetic Mott-insulator with a particular focus on the lower Hubbard band (LHB). The characteristic temperature for 1T-TaSe_2 bulk crystals is below 200 K, whereas the authors observe no comparable temperature dependence for the system investigated here - at least not in the experimentally accessible temperature range. From the smooth changes observed, the authors extrapolate a characteristic temperature of ~ 530 K. Such monolayersystems have already been investigated, by photoemission spectroscopy and STM.

The spectral features (here particularly the LHB) reveal a surprising robustness against several external parameters, as temperature, electron doping by potassium deposition, or even photoexcitation with a 1.55 eV pump-laser. This robustness is in the focus of this work and has not been investigated in earlier works in this detail.

Although the complementary experimental approaches and their results are carefully discussed in a common physical frame, there remains the possibility that the observed robustness is the consequence of a much more trivial effect, namely a structural modification of the monolayer due to an interaction with the substrate. Such an elastic 2D lattice distortion might be fully unrelated the CDW-Mott picture being discussed here. Although alternative explanations for the origin of the gap are discussed briefly in the Supplementary Notes, I can not see how this possibility can be ruled out. As being stated, the interlayer coupling in bulk crystals is important for their physical properties, but here the coupling between graphene substrate and TaSe₂ has not been elucidated.

In contrast to earlier publications (by this and other groups), a novel and strong experimental feature of the present work is the investigation of the temperature dependence of the spectra. Unfortunately, the maximum temperature of the experiment is 450 K, clearly below the expected transition. However, there is a broadening of the so-called LHB which shifts its leading edge seemingly towards the Fermi level. This analysis (or interpretation) should be reconsidered after another fitting of the spectra which considers in particular thermal broadening of the line. The direct comparison with the BCS gap is actually misleading. Furthermore, I was wondering whether the Ta4f core-level spectra are showing a temperature dependence, since their splitting indicates the (temperature dependent) charge distribution within the star-of-David distortion. There might be more information in the temperature dependence which possibly supports the conclusions of this manuscript, namely that the observed robustness is an intrinsic property of the monolayer.

Reviewer #3 (Remarks to the Author):

Authors studied how the CDW-Mott gap size changes when TaSe₂ becomes a single layer. They presented a comprehensive ARPES study on epitaxially grown monolayer 1T-TaSe₂ and 1T-NbSe₂. They further showed how the ARPES data changes under heating, electron doping, and photoexcitation. The transition temperature is found ~530K, making this observation interesting.

In principle the main observation presented in the manuscript is interesting. However, there are some relevant information and discussion are missing, and it is difficult for me to make a judgement based on the current data. I would like to ask authors to present:

1. how the samples were prepared; how monolayer samples were achieved and their STM topographic image or atomically resolved HAADF-STEM image to see how clean the sample is; any surface effects.

2. Is it possible to obtain transport data and compare them with bulk samples? Is there a way to prove that it is indeed a Mott insulator without a magnetic ordering. If yes, please present them. If not, difficulties/challenges in obtaining such data and their implications (e.g., lack of proof on the absence of magnetic ordering) on the final conclusion need to be discussed.

3. The Hubbard interaction U is a local atomic interaction. It is not clear how the gap is controlled by U/W . A simple argument like the ratio U/W change is not convincing in low dimensional system. Indeed W is going to be smaller, since the interlayer hopping is blocked by making a system to be two dimensional. But how does it change a local U size? It is likely that the screening effect is important to understand the increase of U (which in turn increases gap and T_c). Authors need to give some convincing arguments (not a full theory, as it is not a theory paper) on how the interaction strength can be renormalized by the hopping processes allowed in 2D vs. 3D.

Response to the comments from Reviewers

Dear Reviewers;

We greatly appreciate the thoughtful and constructive comments from all the reviewers. We are also deeply grateful to the reviewers for their recognition of the importance of our work by writing that, “*The topic is of large interest to a wide scientific community and the methods are appropriate.*” (Reviewer #1), “*the complementary experimental approaches and their results are carefully discussed*” (Reviewer #2), and “*the main observation presented in the manuscript is interesting*” (Reviewer #3). We have significantly revised the manuscript by fully incorporating all the suggestions from the reviewers. In particular, to strengthen our main claim, we have additionally performed temperature-dependent core-level spectroscopy and low-temperature STM experiments, and verified the crystal structure, its monolayer nature, and the CDW formation with the $\sqrt{13}\times\sqrt{13}$ reconstruction. We have included all these results in the revised manuscript. We present our responses to the respective reviewers and elaborate on how we have revised the manuscript to resolve their concerns, as detailed in the following. The reviewers’ comments are shown in ***bold italic***.

We thank again all the reviewers for their useful and constructive comments to improve our manuscript. We believe that the manuscript has been appropriately revised and is now suitable for publication in Nature Communications.

Sincerely,
Takafumi Sato
WPI-AIMR and Department of Physics,
Tohoku University

P.S.

We have added Hirofumi Oka and Tomoteru Fukumura in the author list, because they have greatly contributed to the additional STM experiments. All co-authors agree with this change.

To the comments from Reviewer #1:

Reviewer comment: *The manuscript of Nakata et al. discuss the robustness of charge-density-wave/Mott phases in monolayer (ML) transition metal dichalcogenides (TMDs). The topic is of large interest to a wide scientific community and the methods are appropriate. There are, however, several points that require additional clarifications before granting publication to Nat. Comm.*

Our response: We thank Reviewer #1 for spending his/her precious time to carefully read our manuscript and giving several constructive suggestions to improve our manuscript. Following the useful suggestion, we have incorporated additional clarifications in the revised manuscript, by carrying out additional STM and core-level spectroscopy experiments to strengthen our main claim regarding the crystal structure, its monolayer nature, and the CDW formation in our 1T-TaSe₂ film, as detailed below. Our point-by-point response to the reviewer’s comments is as follows:

Reviewer comment: - *Samples - Concerning the ML specimens: have the authors performed any LEED / STM / AFM analysis on “these” specific samples (similar to what is reported in Ref. [17-18]) in order to ensure the formation of 1T phase and the sqrt(13x13) lattice reconstruction? (in particular, NbSe2 is very likely to grow as 2H). Furthermore, is there any evidence that TaSe2 and NbSe2 are truly ML? A bilayer system would definitely change their conclusions about the intra-layer vs inter-layer hopping. I believe some stronger proofs should be given other than the ARPES comparison reported in SI note 5.*

Our response: We totally agree with the reviewer that a stronger proof of the 1T phase, $\sqrt{13}\times\sqrt{13}$ lattice reconstruction, and monolayer nature should be necessary. To this end, we have additionally performed STM measurements on our film, in collaboration with the STM group. As seen from the obtained STM image in a 100×100 nm² spatial region for our TaSe₂ film at $T = 4.8$ K

Fig. R1: **a**, STM image in a surface area of 100×100 nm² for monolayer 1T-TaSe₂ on bilayer graphene measured at $T = 4.8$ K. **b**, Height profile along a cut crossing a step of TaSe₂ island shown by red arrow in (a). **c**, STM image in a surface area of 8×8 nm², together with the unit cells of original lattice (orange rhombus) and $\sqrt{13}\times\sqrt{13}R13.9^\circ$ lattice (green rhombus). **d**, Fourier-transform image of (c). **e**, Typical dI/dV curve on the TaSe₂ island measured at $T = 4.8$ K.

in Fig. R1a, a few triangular TaSe₂ islands (yellow region) are recognized on top of bilayer graphene substrate (dark region). We have also confirmed from the height profile along a cut across a step of TaSe₂ island in Fig. R1b (obtained along red arrow in Fig. R1a) that the step height is ~0.94 nm, which is in between monolayer (0.63 Å) and bilayer (1.26 Å) heights in bulk TaSe₂ [Wilson *et al.*, Adv. Phys., **24**, 117–201 (1975); new ref. 1 in Supplementary information], supporting the monolayer nature of our film. High-resolution STM image in Fig. R1c obtained in a narrower spatial region of 8×8 nm² signifies a periodic modulation associated with the hexagonal lattice of star-of-David clusters. We have confirmed that this lattice has a periodicity of $\sqrt{13}\times\sqrt{13}R13.9^\circ$ expected for the formation of star-of-David clusters, which are visible as super-spots in the Fourier transform image shown in Fig. R1d, in good agreement with the STM image of monolayer 1T-TaSe₂ reported in ref. 20 (ref. 19 in the previous manuscript). The monolayer nature is also consistent with our experience on the fabrication of various monolayer TMD films (such as NbSe₂, VTe₂, VSe₂, and TiSe₂) where the monolayer film is formed immediately after the disappearance of buffer-layer-originated $6\sqrt{3}\times 6\sqrt{3}$ streak pattern in the RHEED pattern upon co-evaporation of transition-metal and chalcogen atoms [e.g. refs. 18, 19; Umemoto *et al.*, Nano Res. **12**, 165 (2019) (new ref. 53); Sugawara *et al.*, Phys. Rev. B **99**, 241404(R) (2019) (new ref. 54)]. Although it is hard to directly conclude the formation of the 1T phase but not the 1H phase from the STM image itself because the bottom Se layer is hardly resolved, the former is strongly suggested from the fact that previous studies on bulk TMDs have shown that the star-of-David clusters appear only in the 1T phase. As for NbSe₂ films, we have already established the growth-condition to selectively fabricate monolayer 1H and/or 1T phase by accurately controlling the substrate temperature [ref. 19; Y. Nakata *et al.*, npj 2D Mater. Appl. **2**, 12 (2018), cited as new ref. 55]. In this study, we followed the same recipe to grow monolayer 1T-NbSe₂, since the 1T structure, the $\sqrt{13} \times \sqrt{13}$ lattice reconstruction, and the monolayer nature were already clarified in the previous study. We have elaborated on these points in the revised manuscript (p. 4, lines 82-84; p. 4, line 104 – p. 5, line 110; Methods, p. 14, lines 331-339, p. 15, lines 366-370; Supplementary note 1) by also incorporating Figs. R1a, R1b, R1c, R1d, and R1e, as Figs. S1a, S1b, Fig. 1d, Fig. 1e, and S3, respectively.

Reviewer comment: - Time-resolved ARPES - At line 150 the authors claim a 5.2 work function for TaSe₂ and thus a binding energy range of 1 eV (given the 6.2 eV photons), but Fig. 2f-2g are not in agreement with such statement.

At line 164-165 they also claim that CDW-Mott phase is hardly melted, in contrast to bulk TaS₂. However, this fact depends critically on the pump fluence (they declare 0.26 mJ/cm² in Fig.2) and experimental conditions. There are several experimental evidences (see f.i. recent Dragan Mihailovic's works) showing that to trigger (electronic) phase transitions in Ta-based TMDs a pump fluence of the order of 1 mJ/cm² is required. Thus, it is not surprising that the authors do

not see any "melting" of CDW-Mott phase. In order to demonstrate robustness against photo-excitation a more extensive investigation is necessary. Besides, in SI note 6, the authors report an estimate of the absorbed energy/pulse based on the optical properties of bulk TaS₂. Although optical data and formulas should be specified, a ML film might behave very differently from a bulk and the substrate underneath might play a relevant role in the optical response. Finally, the fluence comparison with Ref.[32] is not the most appropriate: see f.i. recent works of Kapteyn-Murnane group (Sci. Adv. 5, eaav4449 (2019), PNAS 117 8788 (2020)) dealing with TaSe₂.

Considering that (i) ARPES with 6.2 eV photons (Fig. 3f) does not bring substantial additional information compared to He lamp ARPES and that (ii) the time-resolved experiment is not truly necessary for the goal of this work, I suggest to remove this part and maybe address the time-resolved features in a different publication.

Our response: We understand that the reviewer's comment of "Fig. 2f-2g are not in agreement with such a statement" means that "it is unphysical to observe a finite photoelectron signal at the binding energy higher than 1 eV". As mentioned by the reviewer, we have estimated the upper limit of binding energy to be ~ 1 eV from the difference between the photon energy ($h\nu = 6.2$ eV) and the expected work function (5.2 eV). This suggests that one can trust the photoelectron signal up to ~ 1 eV from E_F , but it does not necessarily mean that a photoelectron signal is not detected in the energy region higher than 1 eV. Such a spurious photoelectron signal is known to appear at very low kinetic energy, because the low kinetic energy cut-off of photoelectron signal is usually broad unless a bias voltage is applied between the sample and analyzer. Also, it is known that photoelectrons with kinetic energy typically lower than a half of pass energy of analyzer cannot be correctly measured by a standard ARPES apparatus. In the revised manuscript, we have added a few sentences to describe this point (p. 10, lines 1-7 from the bottom of Supplementary note 7).

As pointed out by the reviewer, one needs to be well aware of the pump fluence when discussing the melting of CDW-Mott phase. Considering the high pump fluence to trigger the electronic phase transition in bulk Ta-based TMDs, we tried to increase the pump fluence as much as possible, and found that 0.26 mJ/cm² is an upper limit of reliable measurements, above which an irreversible spectral broadening was observed due to the photo-induced damage of monolayer sample. Therefore, we carried out the tr-ARPES measurement with 0.26 mJ/cm² pump fluence. On the other hand, it is true that there exist some reports that bulk samples are still robust at this pump fluence because the pump fluence higher than 1 mJ/cm² can be applied without damaging the sample, as pointed out by the reviewer. This suggests that the monolayer sample is structurally and/or chemically more fragile against the pump laser irradiation than the bulk counterpart. This may be related to the absence of interlayer coupling in monolayer which perhaps helps strengthen the overall sample stability. Thus, direct comparison of the robustness of CDW-Mott phase against photo-excitations between monolayer and bulk under identical experimental condition would be difficult, as pointed out by the reviewer.

Nevertheless, we think that, aside from a comparison with the bulk, it would be still meaningful to comment on the experimental fact that the CDW-Mott phase could not be melted in monolayer even when we adopted the maximum pump fluence above which monolayer samples were damaged. We think that this information would be useful for readers, in particular, researchers who aim to carry out pump-probe experiments (including tr-ARPES experiments) of monolayer and ultrathin TMDs, so that we would like to keep the tr-ARPES data in the manuscript. Since we also agree with the reviewer that the tr-ARPES data cannot be used as a direct proof of robust Mott gap against photo excitation, we have toned down our discussion on this matter by moving the tr-ARPES data to Supplementary note. Also, we have explicitly stated the issue of pump fluence described above (p. 7, lines 163-166; Supplementary note 7), by citing a few works by Dragan Mihailovic and Kapteyn-Murnane groups as new refs. 34-36 in the main text and new refs. 18-20 in Supplementary information.

Reviewer comment: - Potassium doping - *At line 172-173 the potassium dose is given in terms of "deposition time" which is not particularly useful (it depends on the experimental conditions) and very qualitative. The relevant information is the K coverage in ML. The authors should at least provide an estimate of the dose to allow a quantitative analysis of the electronic doping. As an example: is the K coverage (atoms/cm²) comparable to the "Stars of David" density (roughly $7 \times 10^{13} \text{ cm}^{-2}$ in TaSe₂)?*

Our response: We thank the reviewer for this important suggestion. We have calibrated the deposition rate of K atoms by calculating the volume of π -band Fermi surface at the K point in bilayer graphene on SiC with keeping the same evaporation rate as that in the case of monolayer TaSe₂, and it is estimated to be $1.6 \times 10^{13} \text{ atoms/cm}^2/\text{min}$. Therefore, $T_d = 2 \text{ min}$ (4 min) in Fig. 3 corresponds to the K coverage d_K of 3.2 (6.4) $\times 10^{13} \text{ atoms/cm}^2$, i.e. $\sim 50 \%$ ($\sim 100 \%$) of the Star-of-David density. Thus, the amount of K dosing with respect to the Star-of-David density is sufficient to achieve a sizable electron doping into monolayer 1T-TaSe₂. In the revised manuscript, we have added a few sentences to explain the calibration of deposition rate (Methods, p. 14, lines 339-345), and replaced the deposition time T_d with the K coverage d_K throughout the manuscript and Figures (Figs. 3b, 3c, 3e, 3f, 3g, and 3h).

Reviewer comment: - Hubbard bands -

Referring to the discussion starting at line 178 (and related Fig.3d-f), the LHB is about 250 meV below E_F in the pristine sample. This means that the UHB should be roughly 250meV above E_F (as properly stated in SI note 2). After 4 min K deposition, the LHB shifts to 750 meV below E_F . Therefore, the UHB should be well below E_F . Authors claim that the spectral weight appearing close to E_F in Fig. 3f might be the tail of the UHB (line 191). However, I strongly doubt this could be possible: the fully populated LHB is actually a half-filled band (1 electron per site). Any hopping

event would lead to double occupancy (e-e at one site and h-h at a neighboring site). This excited state has a Coulomb cost (U) that ends up in populating the UHB. I cannot imagine a strongly or even fully populated UHB below E_F . Instead, electron doping (due to K deposition) would most likely turn the (half-filled) Hubbard band into a standard (fully occupied) metallic band, which is energetically more convenient. The spectral weight at E_F could be simply due to potassium (as more reasonably stated in line 192).

Our response: We thank the reviewer for this insightful suggestion. The energy position of LHB shifts from 0.28 eV to 0.75 eV with K deposition. If the energy bands shift in a rigid-band manner, one would expect the UHB to be located well below E_F in the high K coverage ($d_K = 6.4 \times 10^{13}$ atoms/cm²) sample since the full Mott-gap is estimated to be 0.5 eV (this value is obtained from the tunneling spectrum shown in Fig. R1e). In this regard, as pointed out by the reviewer, it would be difficult to satisfactorily explain the observed finite density of states at the Fermi level (E_F) in terms of the electron population in the UHB. Most likely, the LHB and UHB partially melt upon K doping and a metallic state (which is a mixed state of K and Ta) starts to emerge, although the melting does not proceed so rapidly as that in bulk. In the revised manuscript, we have toned down our discussion about the possible observation of UHB, by putting more emphasis on the other possibilities. Also, we have stated that it would be difficult to fully or strongly populate electrons into the UHB due to the Coulomb cost (p. 8, lines 195-205).

Reviewer comment: - *Doping comparison - See lines 202-204: there is a crucial difference between electron doping via alkali deposition and electron doping in a bulk, such as reported in Ref.[37-38]. In the first case the doping is surface-confined, it leads to space charge layer, band bending and so forth. In the second case the doping is intrinsically induced during the growth process and has no depth dependence (no space charge layer, no band bending). So, these are different physical conditions leading to different physical properties. In particular, the potassium-induced shift of the binding energy seen with ARPES (Fig. 3 d-f) points at the formation of a space charge layer with a downward band bending (as expected for an electron donor doping such as K, see f.i. the work on black phosphorus by Kim et al. Science 349, 723 (2015)). This fact casts some doubts on the true ML nature of the TMD film and on the interlayer hopping argument (see my first comment about the samples). Authors should clarify this point.*

Our response: As mentioned above, our STM measurements signified the 1ML nature of our TaSe₂ film. Our result suggests that the K deposition (~50 % / ~100 % of the start-of-David density) dopes electron carriers into a whole monolayer film and causes an intrinsic change in the band structure within the whole film. Such type of electron doping was identified in other 1ML TMD films such as 1ML TiSe₂ where K deposition causes an overall downward shift of the band structure and disappearance of CDW [S. Kolekar et al., Adv. Quantum Technol. 1, 1800070 (2020), cited as new

ref. 37]. Thus, we think that it is difficult to explain the observed K-deposition-induced shift of LHB in terms of a space charge layer with the downward band bending. We have added a sentence to explain the characteristics of electron doping in our film (p. 7, lines 173-175).

Reviewer comment: - Additional comments – line 28: “...up to high temperature.”. A more specific statement is required.

Our response: We changed “up to high temperature” to “well above the room temperature” because the onset of estimated CDW-Mott transition temperature is 530 K (p. 1, line 32).

Reviewer comment: line 51: “... U/W exceed the critical value”. A Reference is required.

Our response: The charge-transfer gap of cuprates is about 1.4-2.0 eV according to the literature by Ohta *et al.* [Phys. Rev. Lett. **66**, 1228 (1995)]. Since $W = 0.4$ eV, the effective U/W value becomes 3.5-5.0, much larger than the critical value (~ 1). We have stated this point by citing the work by Ohta *et al.* as new ref. 9 (p. 2, lines 52-54).

Reviewer comment: lines 114-117: “Such spectral features...” The absence of Mott gap and the persistence of CDW phase was seen f.i. in Ref.[27] at room temperature. Authors should clarify.

Our response: As pointed out by the referee, bulk TaSe₂ (ref. 27) shows the absence of Mott gap and the persistence of CDW at room temperature. In bulk TaSe₂, the LHB essentially vanishes at room temperature and a large metallic spectral weight emerges at E_F , in contrast to the low temperature (70-220 K) data that displays a peak associated with the LHB. Our ARPES data for 1ML 1T-TaSe₂ at room temperature and $T = 450$ K resembles that of bulk TaSe₂ at low temperature (Fig. 2d), suggestive of the persistence of a Mott gap at $T = 450$ K. We have rephrased the ambiguous statements in the previous manuscript “because the spectral feature continuously evolves in the temperature range of 40-450 K” and stated more explicitly the spectral similarity between the high-temperature data in monolayer TaSe₂ and the low-temperature data in bulk TaSe₂ (p. 5, lines 125-131).

Reviewer comment: lines 125-126: “We expect...”: what do authors mean by “transport gap” and “spectroscopic gap minimum Δ_{Mott} ”? Data in Fig.2e seem to follow a BCS-like behavior, which normally is attributed to the CDW gap (in fact the model in Ref.[26] refers to CDW), but these data refer to the leading edge at Gamma that is the LHB. This is confusing. Authors should clarify.

Our response: There exist two types of definitions in estimating the gap size from the EDC, i.e. (i) the energy position of peak relative to E_F and (ii) the energy position of leading-edge midpoint (LEM) relative to E_F . Both definitions are often used to discuss the gap size in other systems such as cuprates and Fe-based superconductors [e.g. Ding *et al.*, Nature **382**, 51 (1996); Kanigel *et al.*, Nat. Phys. **2**,

447 (2006), cited as new refs. 10 and 11 in Supplementary information]. In the present study, they are named Δ_{Mott} and Δ_{LEM} , respectively. We call Δ_{Mott} a spectroscopic gap, because $2\Delta_{\text{Mott}}$ spans the energy positions at which the LHB and UHB take the highest (and thereby spectroscopically prominent) density of states (DOS). The DOS for both LHB and UHB is not so sharp and has a broad tail as seen from the ARPES data in Fig. 2d and tunneling spectrum in Fig. R1e. Since the thermal excitation starts as soon as the excitation energy exceeds the zero DOS range around E_{F} , the transport measurement is sensitive to this tail. Since the definition of Δ_{LEM} inherently includes the tail region of LHB, Δ_{LEM} is regarded to be sensitive to the transport gap (*i.e.* an activation gap in transport measurements). We have elaborated on these points in the revised manuscript (p. 6, lines 140-141) by adding Supplementary note 5.

Reviewer comment: lines 129-130: "This is also true..." what is also true? the linear behavior (I guess)? or the finite value at high temperature? Sentence is misleading.

Our response: We apologize to the reviewer for this misleading sentence. We intended to mean that the nearly linear behavior as a function of T is also seen in bulk 1T-TaS₂ near $T_{\text{CDW-Mott}}$, as pointed out by the reviewer. We have revised this misleading sentence in the revised manuscript (p. 6, lines 143-145).

Reviewer comment: line 132: authors should report the equations.

Our response: As written in Supplementary note 4 of the previous manuscript, the BCS gap equation is $\Delta_{\text{LEM}}(T)^2 - \Delta_{\text{LEM}}(T_{\text{CDW-Mott}})^2 \propto \tanh^2(A\sqrt{(T_{\text{CDW-Mott}}/T)-1})$, where Δ_{LEM} , $T_{\text{CDW-Mott}}$, and A are the binding energy of leading-edge midpoint, the CDW-Mott transition temperature, and the proportional constant, respectively. In the revised manuscript, we have added this equation and associated explanation in the main text (p. 6, lines 145-149; caption to Fig. 2).

Reviewer comment: line 198 and Fig. 3f: I find this comparison misleading and too heterogeneous to be significant. There are data extrapolated from different experiments, different authors on different samples. I strongly suggest to reconsider the use of this Figure.

Our response: The $I_{\text{EF}}/I_{\text{LHB}}$ values for photoexcited bulk 1T-TaS₂, bulk 1T-TaS₂ ($T = 300$ K and 30 K), and monolayer 1T-TaSe₂ were referenced from ref. 31 (ref. 28 in the previous manuscript), obtained in the present study with a He lamp, and obtained with synchrotron radiation, respectively. Therefore, the experimental condition is different among these measurements. In this regard, we agree with the reviewer that the $I_{\text{EF}}/I_{\text{LHB}}$ values obtained with different experimental set up should not be directly compared with each other unless one can quantitatively estimate the influence of experimental-condition difference. On the other hand, it would be still meaningful to compare the $I_{\text{EF}}/I_{\text{LHB}}$ values when a comparison among different samples are made with the same experimental

condition. As shown in Fig. R2, I_{EF}/I_{LHB} values at $T = 300$ K for pristine monolayer 1T-TaSe₂ obtained with He lamp (green circle) and synchrotron radiation (red circle) show a reasonable agreement within experimental uncertainty ($I_{EF}/I_{LHB} = 0.10 \pm 0.02$ for He lamp and 0.13 ± 0.02 for synchrotron). Thus, a comparison of I_{EF}/I_{LHB} between bulk 1T-TaS₂ and monolayer 1T-TaSe₂ would be still meaningful, and it further supports the intrinsic nature of spectral-weight difference between bulk and monolayer. On the other hand, it would be better to omit the data for photoexcited bulk 1T-TaSe₂ (ref. 31; ref. 28 in the previous manuscript), because it was measured with a different experimental condition. In the revised manuscript, we have elaborated on these points (p. 8, lines 190-194) by replacing old Fig. 3h with Fig. R2.

Fig. R2: Plots of intensity at E_F with respect to that at LHB, I_{EF}/I_{LHB} , as a function of K coverage d_K , estimated from the EDCs in Fig. 3g and Fig. 2d. The values for bulk TaS₂ measured at $T = 30$ and 300 K are also plotted.

Reviewer comment: line 219: "... possible ferromagnetism was not successful". Authors should specify what kind of attempt they made.

Our response: Since the detection of ferromagnetism by macroscopic magnetization measurements like SQUID measurement is difficult in 1ML sample, we have carried out a very primitive experiment to detect possible ferromagnetism, *i.e.* just putting a strong Nd magnet (magnetic field ~ 500 mT) on top of a thin film to detect possible attractive force. But such an attempt has not been successful. We have described this point in the revised manuscript (p. 9, lines 225-228).

Reviewer comment: line 237: "...between monolayer and bulk." A Reference is required.

Our response: The U value for bulk 1T-TaS₂ is ~ 0.7 eV according to ref. 16 (ref. 15 in the previous manuscript). We have cited ref. 16 in the revised manuscript (p. 11, line 269).

Reviewer comment: line 257: "...previous ARPES studies above T_{Mott} ...". A Reference is required.

Our response: We have added references on the previous ARPES studies of bulk TaS₂ (p. 12, line 1; new refs. 21, 26, 27 and 41).

Reviewer comment: line 273: "... a smaller interlayer hopping...". This is not clear: authors are arguing on the Mott and CDW gaps of the ML systems and infer about the magnitude of the interlayer hopping (which in a ML cannot exist!)

Our response: "Interlayer hopping" is a mistype of "intralayer hopping". We have corrected it (p. 12, line 295).

Reviewer comment: line 278: "The stronger CDW...": in what sense the CDW is stronger? Do authors refer to periodic lattice distortion?

Our response: Yes, the stronger CDW means the stronger periodic lattice distortion. We have corrected it (p. 12, lines 300-302).

Reviewer comment: line 320: What is the time resolution (or UV pulse duration) of the time-resolved ARPES setup?

Our response: Time resolution for the KBBF-based time-resolved ARPES is 480 fs at photon energy of 6.2 eV. We have indicated it by adding new ref. 56 (Methods, p. 15, line 355).

Reviewer comment: Fig. 1d, 2d, 2i, 3g, 4c: y-axis should report numbers (even if a.u. are used).

Our response: Following the reviewer's suggestion, we have added numbers in the y-axis in Figs. 1f (Fig. 1d in the previous manuscript), 2d, 3g, and 4c, and Fig. S6d (Fig. 2i in the previous manuscript).

Reviewer comment: Fig. 2e, 3h, 4f-g: all experimental dots should have corresponding error bars.

Our response: We have included error bars in Figs. 2e, 3h, and 4f, and 4g.

To the comments from Reviewer #2:

Reviewer comment: Sato et al. present a detailed experimental study on 1T-TaSe₂ and 1T-NbSe₂ being prepared as monolayers epitaxially grown on bilayer graphene. The graphene itself was prepared on SiC(0001) single crystals.

In the photoemission data the authors observe characteristic spectroscopic differences compared to respective the bulk material, in particular in the temperature dependence of the features in the valence band. The observations are thoroughly discussed in the well established picture of for the non-magnetic Mott-insulator with a particular focus on the lower Hubbard band (LHB). The characteristic temperature for 1T-TaSe₂ bulk crystals is below 200 K, whereas the authors observe no comparable temperature dependence for the system investigated here - at least not in the experimentally accessible temperature range. From the smooth changes observed, the authors extrapolate a characteristic temperature of ~530 K. Such monolayer systems have already been investigated, by photoemission spectroscopy and STM.

The spectral features (here particularly the LHB) reveal a surprising robustness against several external parameters, as temperature, electron doping by potassium deposition, or even photoexcitation with a 1.55 eV pump-laser. This robustness is in the focus of this work and has not been investigated in earlier works in this detail.

Although the complementary experimental approaches and their results are carefully discussed in a common physical frame, there remains the possibility that the observed robustness is the consequence of a much more trivial effect, namely a structural modification of the monolayer due to an interaction with the substrate. Such an elastic 2D lattice distortion might be fully unrelated to the CDW-Mott picture being discussed here. Although alternative explanations for the origin of the gap are discussed briefly in the Supplementary Notes, I can not see how this possibility can be ruled out. As being stated, the interlayer coupling in bulk crystals is important for their physical properties, but here the coupling between graphene substrate and TaSe₂ has not been elucidated.

Our response: We thank Reviewer #2 for spending his/her precious time to carefully read our manuscript and giving us many insightful suggestions to improve our manuscript. We are very thankful to hear that the reviewer recognizes the importance of our work by writing “complementary experimental approaches and their results are carefully discussed”. We also agree with the reviewer that trivial effects such as the interaction with the substrate must be excluded to account for the robust gap in monolayer 1T-TaSe₂ and NbSe₂. We think that the substrate effect is unlikely to be responsible for the robust gap because of the following reasons. If the robust gap opens due to the interaction with the substrate, one may expect that the LHB of TaSe₂/NbSe₂ is hybridized with the graphene bands through a direct band overlap. However, this band hybridization would not occur around the Γ point where the LHB exists, because the graphene band is located 4 eV away from E_F around the Γ point. Another possible explanation for the robust gap is the lattice strain by the graphene substrate and resultant change in the band structure. But this is also unlikely because the lattice strain is expected to be weak due to the existence of van der Waals gap between TaSe₂/NbSe₂ and graphene, as suggested by the experimental fact that the in-plane lattice constant estimated from the RHEED pattern in monolayer 1T-TaSe₂ ($a = 3.5 \text{ \AA}$; ref. 18) is similar to that of bulk [$a = 3.47 \text{ \AA}$; Wilson *et al.*, Adv. Phys., **24**, 117–201 (1975), cited as new ref. 1 in Supplementary information]. One may think that the gap opening is due to the moiré potential associated with the lattice mismatch between TaSe₂/NbSe₂ and graphene. But this is also ruled out because the folded subband associated with the moiré potential is not observed. From these arguments, we think that the substrate effect can be safely ruled out from the origin of robust gap. We have elaborated on these points in the revised manuscript (p. 4, lines 89-90; p. 2, lines 2-13 from the bottom of Supplementary note 2).

Reviewer comment: *In contrast to earlier publications (by this and other groups), a novel and strong experimental feature of the present work is the investigation of the temperature dependence*

of the spectra. Unfortunately, the maximum temperature of the experiment is 450 K, clearly below the expected transition. However, there is a broadening of the so-called LHB which shifts its leading edge seemingly towards the Fermi level. This analysis (or interpretation) should be reconsidered after another fitting of the spectra which considers in particular thermal broadening of the line. The direct comparison with the BCS gap is actually misleading. Furthermore, I was wondering whether the Ta4f core-level spectra are showing a temperature dependence, since their splitting indicates the (temperature dependent) charge distribution within the star-of-David distortion. There might be more information in the temperature dependence which possibly supports the conclusions of this manuscript, namely that the observed robustness is an intrinsic property of the monolayer.

Our response: We agree with the reviewer that the influence from the thermal broadening must be taken into account to reliably analyze the gap size. We have examined whether or not the observed temperature dependence purely originates from the thermal broadening effect. As shown in Fig. R3, we have tried to reproduce the EDC at $T = 450$ K (red curve) by intentionally broadening the EDC at $T = 40$ K with a gaussian, $\exp(-E^2/2\sigma^2)$, by varying its energy width σ (blue curve). When σ was chosen to match the slope of the leading edge (top curves), the spectral weight around the peak top is significantly overestimated in the simulation. On the other hand, when σ was chosen to match the overall EDC shape relatively well (bottom curves), the simulated curve apparently has a broader leading edge. Also, it is noted that σ values used for the simulation [0.11 (0.18) eV which correspond to $T = 1260$ (2070) K for the top (bottom) EDC] are too large to be reconciled with a simple thermal effect. These results suggest that the observed temperature dependence of EDC can hardly be explained with the thermal broadening scenario. We have elaborated on these points in the revised manuscript by adding Fig. R3 as Fig. S4 (p. 6, lines 150-151; p. 7, line 7 from the bottom – p. 8, line 10 of Supplementary note 4).

Regarding the Ta 4f core-level spectra, we have performed additional temperature-dependent photoemission experiments with synchrotron radiation, and the result is shown in Fig. R4. One can see that the use of synchrotron light ($h\nu = 260$ eV) greatly reduces the spectral background seen in the He II data in previous Fig. 1d, and now the main peaks are much better visualized even without background subtraction. At $T = 40$ K, one can clearly see that the Ta 4f_{5/2} and 4f_{7/2} peaks split into two sub-peaks due to the formation of star-of-David clusters. On elevating temperature, the lower-binding-

Fig. R3: EDC at the Γ point at $T = 450$ K (red curve) for 1ML 1T-TaSe₂ and simulated EDCs (blue curves) that were generated by broadening the experimental EDC at $T = 40$ K with a gaussian $\exp(-E^2/2\sigma^2)$ assuming $\sigma = 0.11$ eV (top) and 0.18 eV (bottom).

energy sub-peak of both $Ta4f_{5/2}$ and $4f_{7/2}$ components is gradually weakened, but the shoulder feature still remains even at $T = 400$ K (note that this is the highest temperature we could reach with the ARPES apparatus in synchrotron). This core-level data is consistent with our conclusion that the Mott phase survives up to $T = 530$ K. We have elaborated on these points (p. 5, lines 111-120) by replacing Fig. 1d (new Fig. 1f) with Fig. R4.

Fig. R4: Temperature-dependence of EDC around the Ta-4f core level measured with $h\nu = 260$ eV for 1ML 1T-TaSe₂.

To the comments from Reviewer #3:

Reviewer comment: *Authors studied how the CDW-Mott gap size changes when TaSe₂ becomes a single layer. They presented a comprehensive ARPES study on epitaxially grown monolayer 1T-TaSe₂ and 1T-NbSe₂. They further showed how the ARPES data changes under heating, electron doping, and photo-excitation. The transition temperature is found ~530K, making this observation interesting.*

In principle the main observation presented in the manuscript is interesting. However, there are some relevant information and discussion are missing, and it is difficult for me to make a judgement based on the current data. I would like to ask authors to present:

Our response: We thank Reviewer #3 for his/her careful reading of our manuscript and giving thoughtful advices and comments to improve our manuscript. To fully respond to the reviewer's request, we have additionally performed STM measurements and characterized our monolayer samples in detail. We have also added several explanations and supplied necessary information to strengthen our main claim. Our point-by-point response to the respective comments from the reviewer is as follows:

Reviewer comment: *1. how the samples were prepared; how monolayer samples were achieved and their STM topographic image or atomically resolved HAADF-STEM image to see how clean the sample is; any surface effects.*

Our response: We totally agree with the reviewer that the characterization of the monolayer samples is important to convince readers of our main claim. As described in the Methods section of the previous manuscript, to grow monolayer 1T-TaSe₂ and 1T-NbSe₂ films, we at first grew bilayer graphene on 6H-SiC, and then co-evaporated Ta (Nb) and Se on the bilayer graphene substrate kept at 560°C (580°C). This as-grown film was subsequently annealed at 400°C for 30 min. The growth process was monitored by the reflection high-energy electron diffraction (RHEED). Based on our experience of

fabricating various monolayer TMD films such as NbSe₂, VTe₂, VSe₂, and TiSe₂ [e.g. refs. 18, 19; Umemoto *et al.*, Nano Res. **12**, 165 (2019) (new ref. 53); Sugawara *et al.*, Phys. Rev. B **99**, 241404(R) (2019) (new ref. 54)], a monolayer film is formed immediately after the disappearance of buffer-layer-originated $6\sqrt{3}\times 6\sqrt{3}$ RHEED pattern upon co-evaporation of transition-metal and chalcogen atoms. We have controlled the thickness of 1T-TaSe₂ and 1T-NbSe₂ films by following this established scheme. As for the monolayer NbSe₂ film, by utilizing the electron diffraction and STM measurements, we have already established the sample-growth condition to selectively fabricate the 1H and/or 1T phases *via* accurately controlling the substrate temperature, as detailed in our previous studies [ref. 19; Y. Nakata *et al.*, npj 2D Mater. Appl. **2**, 12 (2018), cited as new ref. 55]. In this study, we just followed the same recipe to grow monolayer 1T-NbSe₂ established in our previous studies where the 1T structure, $\sqrt{13}\times\sqrt{13}$ lattice reconstruction and its monolayer nature were already clarified. To fabricate a monolayer TaSe₂ film, we also followed the fabrication method established by ourselves (ref. 18). Since the STM characterization of monolayer TaSe₂ film was missing in the previous manuscript, we have additionally performed the STM measurements of monolayer TaSe₂. As seen from the obtained

STM image in a 100×100 nm² spatial region for TaSe₂ film at $T = 4.8$ K in Fig. R5a, a few triangular TaSe₂ islands (yellow region) are recognized on top of bilayer graphene substrate (dark region). We have also confirmed from the height profile along a cut across a step of TaSe₂ island in Fig. R5b (obtained along red line in Fig. R5a) that the step height is ~ 0.94 nm, which is in between the monolayer (0.63 Å) and bilayer (1.26 Å) heights of bulk TaSe₂ [Wilson *et al.*, Adv. Phys., **24**, 117–201 (1975), cited as new ref. 1 in Supplementary information], supporting the monolayer nature of our film. High-resolution STM image in Fig. R5c obtained in a narrower spatial region of 8×8 nm² signifies a periodic modulation associated with the hexagonal lattice of fully stacked star-of-David clusters. We have confirmed

Fig. R5: **a**, STM image in the surface area of 100×100 nm² for monolayer 1T-TaSe₂ on bilayer graphene measured at $T = 4.8$ K. **b**, Height profile along a cut crossing a step of TaSe₂ island shown by red arrow in (a). **c** STM image in the surface area of 8×8 nm², together with the unit cells of original lattice (orange rhombus) and $\sqrt{13}\times\sqrt{13}R13.9^\circ$ lattice (green rhombus). **d**, Fourier-transform image of (c). **e**, Typical dI/dV spectrum on the TaSe₂ island measured at $T = 4.8$ K.

that this lattice has a periodicity of $\sqrt{13}\times\sqrt{13}R_{13.9^\circ}$ expected for the formation of star-of-David clusters, as clearly visible from the location of super-spots in the Fourier transform image shown in Fig. R5d in good agreement with the STM image of monolayer 1T-TaSe₂ reported in ref. 19. This is also consistent with the behavior of RHEED pattern, which shows that a monolayer film is formed immediately after the disappearance of buffer-layer-originated $6\sqrt{3}\times 6\sqrt{3}$ streak pattern upon co-evaporation of transition-metal and chalcogen atoms. Since the STM results show no clear signature for Se defects or other anomalies associated with impurities, crystal imperfections etc., we think that our film is of sufficiently high quality to get rid of extrinsic surface effects. We have elaborated on these points in the revised manuscript (p. 4, lines 82-84; p. 4, line 104 – p. 5, line 110; Methods, p. 14, lines 331-339, p. 15, lines 366-370; Supplementary note 1) by also incorporating Figs. R5a, R5b, R5c, R5d, and R5e, as Figs. S1a, S1b, Fig. 1d, Fig. 1e, and S3, respectively.

Reviewer comment: 2. *Is it possible to obtain transport data and compare them with bulk samples? Is there a way to prove that it is indeed a Mott insulator without a magnetic ordering. If yes, please present them. If not, difficulties/challenges in obtaining such data and their implications (e.g., lack of proof on the absence of magnetic ordering) on the final conclusion need to be discussed.*

Our response: We thank the reviewer for this important suggestion. At the moment, it is difficult to obtain reliable transport data with our monolayer film, because the film is unstable in the atmosphere and therefore not suitable for performing *ex-situ* transport measurements. Even when we cover the sample with a protection layer, the electric current will selectively flow through the metallic bilayer graphene substrate (and the protection layer if it is conductive), and consequently, the insulating behavior of the monolayer film cannot be detected. *In-situ* transport measurements would be also difficult because of the same reason. Even if we find an insulating substrate to grow TaSe₂/NbSe₂ films, there is no guarantee that the grown film shows exactly the same electronic properties as in the case of bilayer graphene substrate. While both *ex-situ* and *in-situ* transport measurements are difficult at the moment, we are able to spectroscopically confirm the insulating nature from the tunneling experiment. As shown in Fig. R5e, the zero DOS is observed in the wide bias voltage range of $\sim\pm 0.05$ eV, a typical signature of insulating behavior. The tunneling spectrum of bulk 1T-TaS₂ in the CCDW-Mott phase in the previous literatures (e.g. ref. 24) shows the zero DOS in the energy range of ~ 0.15 eV, similarly to monolayer TaSe₂. This suggests the insulating ground state of monolayer TaSe₂. Regarding the magnetic ordering, since the detection of magnetism in monolayer film by standard magnetization measurements (such as SQUID) is difficult, it would be necessary to perform the surface sensitive X-ray magnetic circular dichroism (XMCD) measurement with ultrathin films. Since the XMCD experiment is beyond the scope of present study, we would like to leave it as a future challenge. We have elaborated on these points in the revised manuscript (p. 9, line 225 – p. 10, line 231; p. 5, line 13 from the bottom – p. 6, line 4 of Supplementary note 3).

Reviewer comment: 3. *The Hubbard interaction U is a local atomic interaction. It is not clear how the gap is controlled by U/W . A simple argument like the ratio U/W change is not convincing in low dimensional system. Indeed W is going to be smaller, since the interlayer hopping is blocked by making a system to be two dimensional. But how does it change a local U size? It is likely that the screening effect is important to understand the increase of U (which in turn increases gap and T_c). Authors need to give some convincing arguments (not a full theory, as it is not a theory paper) on how the interaction strength can be renormalized by the hopping processes allowed in 2D vs. 3D.*

Our response: We thank the reviewer for this insightful suggestion. It is well-known that the Mott-Hubbard transition occurs when U/W can be tuned above a critical value, where U is the effective on-site Coulomb correlation energy and W is the effective d -electron bandwidth in the material. In the present context of the CDW behavior observed in monolayers of 1T-TaSe₂ and 1T-NbSe₂, it is important to discuss the independent roles of how U and W are independently affected on going from the bulk 3D structure to the monolayer 2D case. The effective on-site Coulomb correlation energy can be described by the equation $U = E_1 - E_A - E_{\text{Pol}}$ where, E_1 is the ionization energy, E_A is the electron affinity, and E_{Pol} is the polarization energy which arises from screening due to any electronic perturbation such as removing or adding an electron. This screening causes a strong reduction of U compared to the bare Coulomb interaction $U_{\text{bare}} (= E_1 - E_A)$. For example, based on a one band Hubbard model, U for the Cu site gets reduced to ~ 4 eV in La₂CuO₄ compared to $U_{\text{bare}} \sim 20$ eV for Cu atom (Werner *et al.*, Phys. Rev. B **91**, 125142 (2015); new ref. 47). For 4d and 5d transition metals, U is expected to be still lower. For 1T-TaS₂, 1T-TaSe₂ and 1T-NbSe₂, typical values of U reported in the literature range from $\sim 0.4/0.7$ eV (in the GW approximation/DFT-DMFT approximation, refs. 16, 51) to $\sim 2.0/2.8$ eV (in GGA + U) [ref. 20 and new ref. 48 (Liu *et al.* Nat. Commun. **12**:1978 (2021))]. Considering the role of screening in monolayer compared to the bulk case, while the intralayer E_{Pol} is expected to show negligible changes in the monolayer case, the interlayer E_{Pol} would be suppressed as there are no other layers and the interaction with the substrate is weak, resulting in an effective increase in U compared to the bulk.

Similarly, since there is no out-of-plane or inter-layer hopping in the monolayer i.e. the intrinsic bulk interlayer bandwidth W_{out} is absent, hence the net effective bandwidth W will get reduced. Reported values of bandwidths for bulk 1T-TaS₂ are $W_{\text{out}} \sim 0.54$ eV, $W_{\text{in}} \sim 0.08$ eV (ref. 50) and $U \sim 0.7$ eV (ref. 16), and although an appropriate value of U is unknown in the monolayer case, the bandwidth change itself would indicate a large increase in U/W on going from bulk to monolayer 1T-TaS₂. Similar changes can be expected for monolayer 1T-TaSe₂ and 1T-NbSe₂. Further, the in-plane bandwidth W_{in} is also expected to reduce due to the structural distortion in the monolayer accompanying the CDW transition. Thus, upon dimensionality reduction from 3D to 2D, both the increase in U and decrease

in W are expected to positively work together to efficiently increase U/W . We have revised the discussion part based on this argument (p. 10, line 242 – p. 11, line 258).

We thank again all the reviewers for their useful and constructive comments to improve our manuscript. We believe that the manuscript has been appropriately revised and is now suitable for publication in Nature Communications.

Sincerely,
Takafumi Sato
WPI-AIMR and Department of Physics,
Tohoku University

REVIEWERS' COMMENTS

Reviewer #1 (Remarks to the Author):

The authors have properly and thoroughly addressed all previous issues. In particular:

- i) they have provided valuable experimental evidence of the monolayer/CDW nature of the TaSe₂ sample with new STM analysis;
- ii) converted potassium deposition time to dose with a proper calibration procedure, that is more meaningful;
- iii) discussed in more detail some ambiguous points of the previous version (the upper Hubbard band issue and Fig. 3f);
- iv) added citations where needed;
- v) the choice to move tr-ARPES data in the supplementary is reasonable.

I believe the manuscript is now eligible for publication.

Reviewer #2 (Remarks to the Author):

I find that the referee's comments and questions have been addressed competently and convincingly by the authors. The reviewed manuscript can be published in its present form.

Reviewer #3 (Remarks to the Author):

Authors have answered the questions that I raised earlier, and added new STM data to further support their conclusion. The manuscript is properly revised consistent with their reply. I recommend the publication.